# Fibroblastic reticular cell-derived lysophosphatidic acid regulates confined intranodal T-cell motility

Akira Takeda[1,2,3], Daichi Kobayashi[1,2], Keita Aoi[2,4], Naoko Sasaki[1], Yuki Sugiura[5,6], Hidemitsu Igarashi[7], Kazuo Tohya[8], Asuka Inoue[9], Erina Hata[1,2], Noriyuki Akahoshi[7], Haruko Hayasaka[1,2], Junichi Kikuta[2,4], Elke Scandella[10], Burkhard Ludewig[10], Satoshi Ishii[7], Junken Aoki[9], Makoto Suematsu[5,11], Masaru Ishii[2,4], Kiyoshi Takeda[2,12], Sirpa Jalkanen[3], Masayuki Miyasaka[1,2,3*†], Eiji Umemoto[1,2*‡]

[1]Laboratory of Immunodynamics, Department of Microbiology and Immunology, Osaka University Graduate School of Medicine, Osaka, Japan; [2]WPI Immunology Frontier Research Center, Osaka University, Osaka, Japan; [3]MediCity Research Laboratory, University of Turku, Turku, Finland; [4]Department of Immunology and Cell Biology, Osaka University Graduate School of Medicine, Osaka, Japan; [5]Department of Biochemistry, Keio University School of Medicine, Tokyo, Japan; [6]JST Precursory Research for Embryonic Science and Technology project, Saitama, Japan; [7]Department of Immunology, Graduate School of Medicine, Akita University, Akita, Japan; [8]Department of Anatomy, Kansai University of Health Sciences, Awaji, Japan; [9]Laboratory of Molecular and Cellular Biochemistry, Graduate School of Pharmaceutical Sciences, Tohoku University, Sendai, Japan; [10]Institute of Immunobiology, Kantonal Hospital St. Gallen, St. Gallen, Switzerland; [11]Core Research for Evolutional Science and Technology project, Saitama, Japan; [12]Laboratory of Immune Regulation, Department of Microbiology and Immunology, Osaka University Graduate School of Medicine, Osaka, Japan

*For correspondence: mmiyasak@ orgctl.med.osaka-u.ac.jp (MM); umemoto@ongene.med.osaka-u. ac.jp (EU)

Present address: †Interdisciplinary Program for Biomedical Sciences, Institute for Academic Initiatives, Osaka University, Osaka, Japan; ‡Laboratory of Immune Regulation, Department of Microbiology and Immunology, Osaka University Graduate School of Medicine, Osaka, Japan

**Abstract** Lymph nodes (LNs) are highly confined environments with a cell-dense three-dimensional meshwork, in which lymphocyte migration is regulated by intracellular contractile proteins. However, the molecular cues directing intranodal cell migration remain poorly characterized. Here we demonstrate that lysophosphatidic acid (LPA) produced by LN fibroblastic reticular cells (FRCs) acts locally to $LPA_2$ to induce T-cell motility. In vivo, either specific ablation of LPA-producing ectoenzyme autotaxin in FRCs or $LPA_2$ deficiency in T cells markedly decreased intranodal T cell motility, and FRC-derived LPA critically affected the $LPA_2$-dependent T-cell motility. In vitro, LPA activated the small GTPase RhoA in T cells and limited T-cell adhesion to the underlying substrate via $LPA_2$. The LPA-$LPA_2$ axis also enhanced T-cell migration through narrow pores in a three-dimensional environment, in a ROCK-myosin II-dependent manner. These results strongly suggest that FRC-derived LPA serves as a cell-extrinsic factor that optimizes T-cell movement through the densely packed LN reticular network.

## Introduction

Blood-borne naïve lymphocytes migrate along the fibroblastic reticular cell (FRC) network in lymph nodes (LNs) (*von Andrian and Mempel, 2003*; *Miyasaka and Tanaka, 2004*; *Girard et al., 2012*). B

**eLife digest** Small organs called lymph nodes are found throughout the body and help to filter out harmful particles and cells. Lymph nodes are packed with different types of immune cells, such as the T-cells that play a number of roles in detecting and destroying bacteria, viruses and other disease-causing microbes. Within the lymph node, T-cells crawl along a meshwork made up of cells called fibroblastic reticular cells. The T-cells appear to move in random patterns, but the signals that drive this movement remain ill-defined.

Now, Takeda et al. reveal that a lipid called lysophosphatidic acid (LPA), which is produced by the fibroblastic reticular cells, is responsible for regulating how T-cells move around inside the lymph nodes. T-cells are able to detect LPA via certain receptor proteins on their surface. Takeda et al. engineered mice that were either unable to produce a particular LPA receptor on their T-cells, or that produced less LPA than normal. The T-cells of these mice moved around less than T-cells in normal mice.

Further experiments revealed that LPA signaling also affects the signaling pathway that alters how well the T-cells stick to nearby surfaces. This suggests that LPA helps to optimize T-cell movement to allow the cells to navigate the small spaces found between the fibroblastic reticular cells. In the future, targeting the processes involved in LPA signaling could help to develop new treatments for disorders of the immune system.

cells then migrate into LN follicles, whereas T cells remain in the paracortex and migrate continually along the FRC network (*Bajénoff et al., 2006*). This intranodal migration provides critical opportunities for T cells to encounter cognate antigen-presenting dendritic cells. Two-photon microscopic analysis has shown that naïve T cells crawl along the FRC network in an apparently random pattern of motion, at an average velocity of 10–15 µm per minute (*Miller et al., 2002*; *Okada and Cyster, 2007*; *Worbs et al., 2007*). FRCs promote intranodal T-cell motility by signaling naïve lymphocytes with CCL21/CCL19 via CCR7, thus activating the small GTPase Rac (*Okada and Cyster, 2007*; *Worbs et al., 2007*; *Faroudi et al., 2010*; *Huang et al., 2007*), although CCR7 signaling only partially account for the interstitial T cell motility (*Okada and Cyster, 2007*; *Huang et al., 2007*).

LPA is a bioactive lysophospholipid produced both extracellularly and intracellularly. Extracellularly produced LPA is involved in such diverse biological functions as vascular remodeling and cell growth, survival, and migration (*Choi et al., 2010*; *Yanagida et al., 2013*). Intracellularly produced LPA is an intermediate in the synthesis of triglycerides and glycerophospholipids, and thought to act as a 'housekeeper' inside the cell (*Mills and Moolenaar, 2003*). Extracellular LPA is predominantly produced by autotaxin (ATX, also referred to as ENPP2 [ectonucleotide pyrophosphatase/phosphodiesterase family member 2]), an ectoenzyme that was originally identified as a tumor-cell motility-enhancing factor (*Stracke et al., 1992*). ATX is a lysophospholipase D that produces LPA by hydrolyzing lysophosphatidylcholine (LPC) (*Okudaira et al., 2010*; *Moolenaar and Perrakis, 2011*). We and others have reported that ATX is strongly expressed in HEV endothelial cells (ECs), and that ATX regulates lymphocyte migration into the LN parenchyma (*Kanda et al., 2008*; *Nakasaki et al., 2008*; *Umemoto et al., 2011*). We also demonstrated that LPA enhances lymphocyte detachment from ECs and promotes lymphocyte transmigration across the HEV basal lamina, at least in part by acting on HEV ECs (*Bai et al., 2013*). LPA also acts on naïve T cells to induce chemokinesis and cell polarization (*Kanda et al., 2008*; *Katakai et al., 2014*; *Zhang et al., 2012*) and transmigration (*Zhang et al., 2012*). While a study using pharmacological inhibitors revealed that ATX/LPA promotes intranodal lymphocyte motility in an ex vivo LN explant model (*Katakai et al., 2014*), the physiological significance of the ATX/LPA axis in interstitial lymphocyte migration remains unknown.

To date, six LPA receptors ($LPA_1$–$LPA_6$) have been identified. LPA receptors couple to multiple G proteins, including $G_i$, $G_{12/13}$, $G_q$, and $G_s$, and upon ligand binding, these G proteins activate diverse intracellular signaling components including Rho and Rac. Although $LPA_2$ has recently been reported to play a role in intranodal T-cell migration (*Knowlden et al., 2014*), it remains unclear how $LPA_2$-mediated signaling affects interstitial T-cell motility and whether LPA is the prime activating ligand.

Leukocyte migration in a confined environment is regulated at least partly by cell contraction (*Lämmermann et al., 2008*). *Jacobelli et al. (2010)* reported that a contractile protein, myosin IIA, is required for T-cell amoeboid motility in confined environments such as LNs (*Jacobelli et al., 2010*). Myosin IIA cross-links actin, thus limiting surface adhesion and allowing T cells to exert contractile force. Myosin IIA's activity is regulated by RhoA and Rho-associated protein kinase (ROCK) signaling. While the cell-extrinsic factor(s) that regulate myosin II's activity during T-cell migration in a confined environment have been poorly defined, a recent study using zebrafish germ progenitor cells showed that LPA induces cell polarization in a ROCK-myosin II-dependent manner, which enables rapid cell migration in a confined environment (*Ruprecht et al., 2015*).

In this study, by using mice conditionally deficient for the LPA-generating enzyme ATX in FRCs and those deficient in LPA$_2$, we demonstrated that bioactive LPA species are produced by FRCs in an ATX-dependent manner and that LPA acts locally on LPA$_2$ on T cells. This LPA$_2$-mediated signaling activates the RhoA-ROCK-myosin II pathway and promotes confinement-optimized interstitial T-cell migration. The FRC-derived LPA thus serves as a cell-extrinsic factor that optimizes T-cell movement through the densely packed LN reticular network, to fine-tune T-cell trafficking.

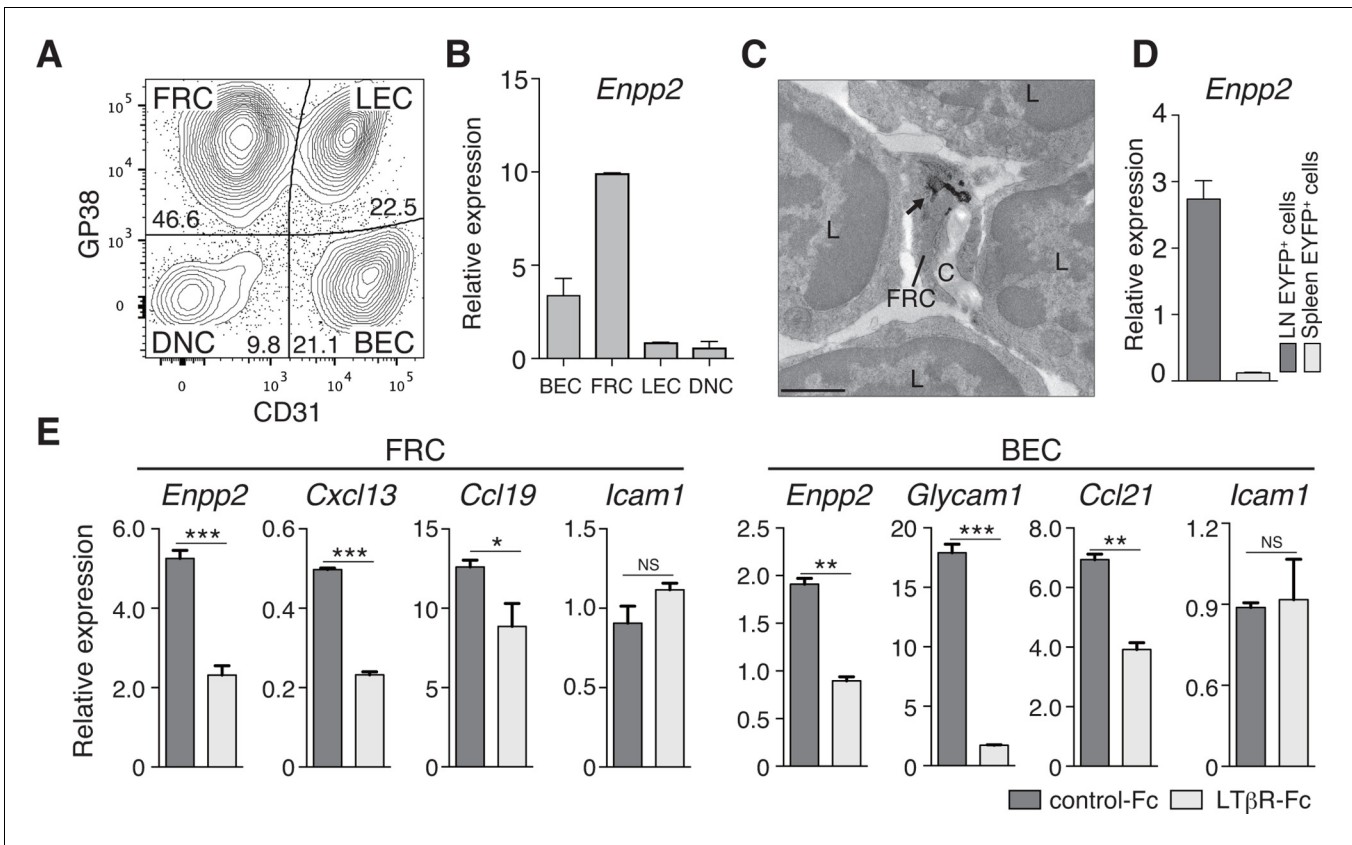

**Figure 1.** FRCs and vascular ECs express autotaxin in an LT$\beta$R-signaling-dependent manner. (**A**) Representative dot plot of the CD45$^-$ stromal cells in LNs. CD45$^-$ cells were divided by CD31 and gp38 expression into FRCs (gp38$^+$ CD31$^-$), lymphatic endothelial cells (LECs; gp38$^+$ CD31$^+$), blood endothelial cells (BEC; gp38$^-$CD31$^+$), and double-negative cells. Numbers indicate the frequency of cells in each gate. (**B**) *Enpp2/Autotaxin* expression in stromal LN fractions, analyzed by quantitative RT-PCR. (**C**) ATX expression examined by immunoelectron microscopy; arrow indicates positive signals in an FRC. C: collagen fibers; L: lymphocytes. Bar, 1 $\mu$m. (**D**) *Enpp2* in CD45$^-$ EYFP$^+$ cells in LNs and the spleen of *Ccl19*-Cre x *R26*-EYFP mice. (**E**) Effect of LT$\beta$R signaling on *Enpp2* expression in FRCs and BECs. LT$\beta$R signaling was blocked by injecting 100 $\mu$g of recombinant LT$\beta$R-Fc or an isotype control into adult mice intraperitoneally weekly for 4 weeks. The FRCs and BECs were then sorted, and the indicated gene expressions were analyzed by quantitative PCR. Data are representative of three (**A, B, D**) or two (**C, E**) independent experiments (n = 3 per group). Differences between groups were evaluated by Student's *t*-test. *$P < 0.05$; **$P < 0.005$; ***$P < 0.0005$.

## Results

### FRCs express the LPA-generating enzyme ATX in a lymphotoxin β receptor (LTβR)-signaling-dependent manner

Although CCR7 ligands are reported to stimulate intranodal T-cell motility (*Okada and Cyster, 2007*; *Worbs et al., 2007*; *Huang et al., 2007*), they are not sufficient to account for effective T-cell migration in LNs. Given that ATX, which generates the motogenic lysophospholipid, LPA, is expressed in HEV ECs to modulate lymphocyte motility (*Nakasaki et al., 2008*; *Bai et al., 2013*), we speculated that ATX and its product LPA are also expressed in other stromal cell subsets in the LN parenchyma and may control intranodal lymphocyte migration. Indeed, a recent paper showed that ATX is expressed in CCL21⁺ CD31⁻ stromal cells in LNs (*Katakai et al., 2014*). We therefore subdivided the CD45⁻ LN stromal cells into four stromal subsets (*Figure 1A*). As shown in *Figure 1B*, *Enpp2/Autotaxin* was readily detected in GP38⁺ CD31⁻ FRCs as well as GP38⁻ CD31⁺ blood endothelial cells (ECs), with negligible expression in lymphatic ECs and double-negative cells. Electron microscopic analysis confirmed that ATX was expressed in the FRCs surrounding collagen fiber bundles (*Figure 1C*). Interestingly, analyses using *Ccl19*-Cre x *R26*-EYFP mice, which constitutively express yellow fluorescent protein (YFP) in FRCs (*Chai et al., 2013*), revealed that *Enpp2* was selectively expressed in LN FRCs but not splenic FRCs (*Figure 1D*). This *Enpp2* expression was apparently dependent on LTβR signaling, because blocking LTβR signaling significantly reduced expression of *Enpp2, Cxcl13*, and *Ccl19*, but not *Icam1*, in FRCs (*Figure 1E*). This blockade also downregulated transcription of HEV marker genes such as *Glycam1* and *Ccl21* in BECs (*Figure 1E*), consistent with a previous report (*Browning et al., 2005*). These results confirm that similarly to HEV ECs, FRCs constitutively express the LPA-generating enzyme ATX, which is maintained at least in part by LTβR signaling.

### Multiple LPA species are produced in the LN parenchyma by FRCs

To verify that LPA is generated in situ by FRC-derived ATX, we crossed *Enpp2^{fl/fl}* mice and *Ccl19*-Cre mice to generate *Ccl19*-Cre *Enpp2^{fl/fl}* mice that lacked ATX expression specifically in the FRCs. As expected, in the *Ccl19*-Cre *Enpp2^{fl/fl}* mice, *Enpp2* was completely lost in the FRCs but not in the BECs, whereas *Ccl21* and GP38 expression was comparable between these strains (*Figure 2A*, *Figure 2—figure supplement 1*). The frequency of FRCs in stromal cells also appeared to be uncompromised by the deficiency of *Enpp2* in FRCs (*Figure 2—figure supplement 1*). We then compared LPA production in the LN of these mice using imaging mass spectrometry (IMS). To this end, we first injected fluorescein-conjugated dextran, which labels lymphatics and the medulla, into the footpad, and LPA (18:0), LPA (18:1), LPA (18:2), and LPA (20:4) were then visualized in LN sections. As shown in *Figure 2B*, signals corresponding to LPA (18:0) were widely distributed in the LN. The signals were comparable in intensity and frequency in *Enpp2^{fl/fl}* and *Ccl19*-Cre *Enpp2^{fl/fl}* mice; this LPA species appears to be produced mainly within the cell (Aoki, J; unpublished observation) independently of ATX (*Yukiura et al., 2011*, *Nishimasu, et al., 2011*). In sharp contrast, signals corresponding to LPA (18:1), LPA (18:2), and LPA (20:4), the major species produced extracellularly by ATX (*Yukiura et al., 2011*), were predominantly observed in the paracortex both close to and at a distance from HEVs, but only marginally in the medulla. These signals were substantially decreased in the cortex of *Ccl19*-Cre *Enpp2^{fl/fl}* as compared with *Enpp2^{fl/fl}* mice (*Figure 2B*).

To verify that the cortical LPA signals associated with non-HEV structures were derived from FRCs, we next mapped the LPA signals relative to HEVs in *Enpp2^{fl/fl}* mice and *Ccl19*-Cre *Enpp2^{fl/fl}* mice by measuring the distance between individual signals and the nearest HEV. As shown in *Figure 2C*, the frequency of LPA (18:1), LPA (18:2), and LPA (20:4) signals within 50 μm of an HEV did not differ significantly between *Enpp2^{fl/fl}* and *Ccl19*-Cre *Enpp2^{fl/fl}* mice. However, the frequency of relatively distant signals (more than 50 μm) decreased substantially when ATX was ablated in FRCs. Hence, the median distance between LPA signals and HEVs was significantly reduced in *Ccl19*-Cre *Enpp2^{fl/fl}* compared with *Enpp2^{fl/fl}* mice, consistent with the idea that distant LPA signals were associated with FRCs. Together with the observations showing robust expression of ATX in FRCs, these findings indicate that the cortical LPA signals not associated with HEVs are mainly produced by FRCs in an ATX-dependent manner.

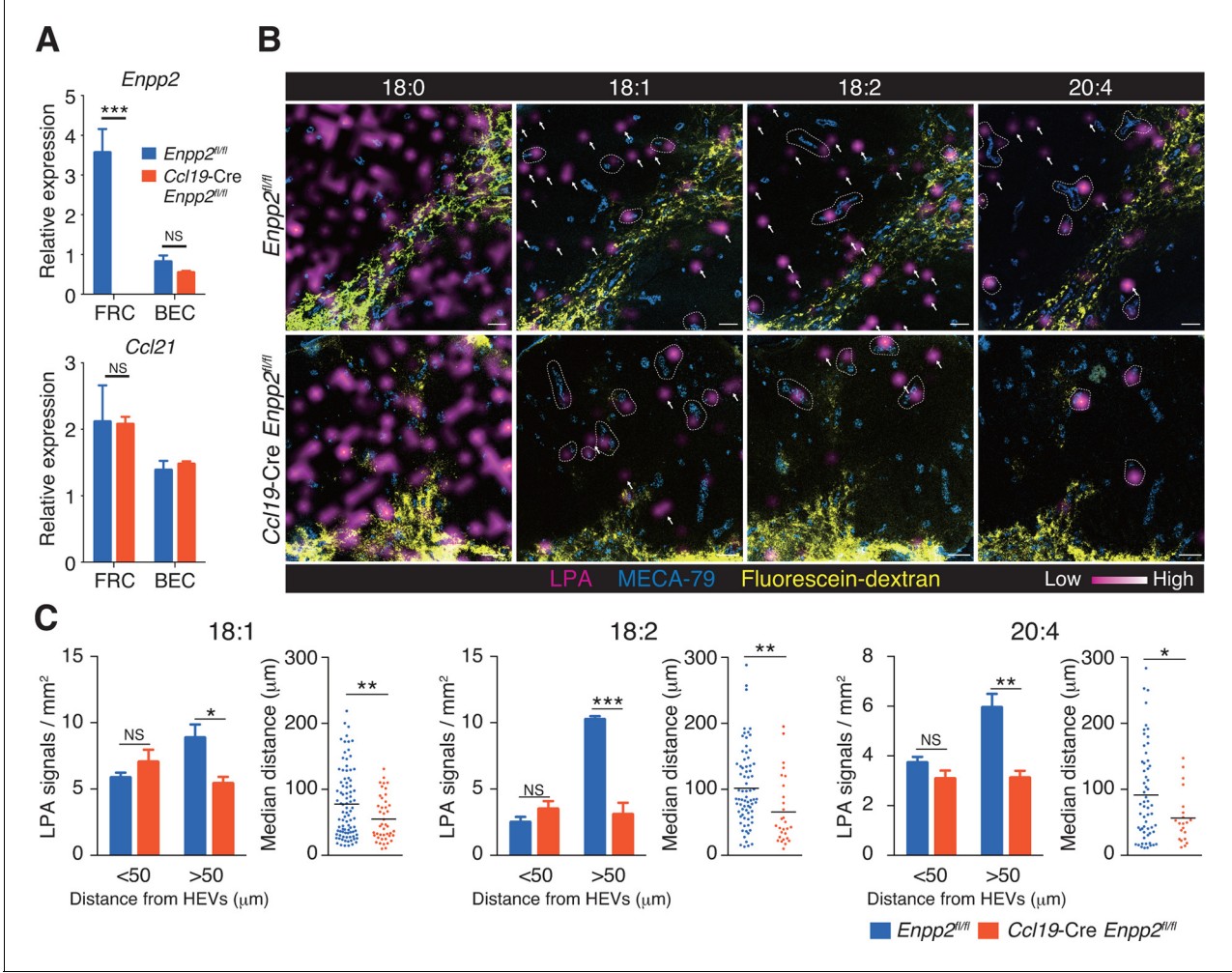

**Figure 2.** Multiple LPA species are produced in the LN parenchyma by FRCs. (**A**) *Enpp2* and *Ccl21* mRNA expression in LN stromal cells in *Ccl19*-Cre *Enpp2*^fl/fl^ or *Enpp2*^fl/fl^ mice. See also analysis of stromal cell populations in those mice in **Figure 2—figure supplement 1**. (**B**) LPA-species distribution in the LNs of *Ccl19*-Cre *Enpp2*^fl/fl^ and *Enpp2*^fl/fl^ mice. Fluorescein-conjugated dextran (pseudocolored in yellow) was injected into the footpads to visualize the lymphatics, and draining LNs were collected 10 min after the injection. Signals corresponding to LPA (18:0) (ion transition from *m/z* 437 to 153), LPA (18:1) (from *m/z* 435 to 153), LPA (18:2) (from *m/z* 433 to 153), and LPA (20:4) (from *m/z* 457 to 153) were visualized by IMS analysis (magenta) and overlapped with MECA-79 staining (blue) on LN serial sections. LPA signals located within 50 μm of an HEV are circled by dotted lines and signals more than 50 μm away from an HEV are indicated by arrows for each of the LPA (18:1), LPA (18:2), and LPA (20:4) species. (**C**) The frequency of LPA signals by distance from an HEV, and the median distance from an HEV. Data are representative of three (**A**) or two (**B**, **C**) independent experiments. Differences between groups were evaluated by Student's *t*-test. \**P* < 0.05, \*\**P* < 0.005, \*\*\**P* < 0.0005. Bars, 100 μm.

The following figure supplement is available for figure 2:

**Figure supplement 1.** Stromal cell populations in *Enpp2*^fl/fl^ and *Ccl19*-Cre *Enpp2*^fl/fl^ mice.

## FRC-derived LPA promotes intranodal T-cell migration

To understand the role of FRC-derived LPA in regulating intranodal T-cell migration, we next examined the CD4^+^ T-cell interstitial migration in LNs by intravital two-photon microscopy. To this end, CD4^+^ T cells from WT mice expressing a transgene encoding enhanced GFP (eGFP) were injected intravenously into *Enpp2*^fl/fl^ and *Ccl19*-Cre *Enpp2*^fl/fl^ mice, and the intranodal T-cell migration in popliteal LNs (PLNs) was imaged 15–25 hr later (**Video 1**). As shown in **Figure 3A, B**, CD4^+^ T-cell movement and displacement from the original location in the PLN was substantially restricted in *Ccl19*-Cre *Enpp2*^fl/fl^ compared with *Enpp2*^fl/fl^ mice. The median T-cell velocity was also lower in *Ccl19*-Cre *Enpp2*^fl/fl^ than in *Enpp2*^fl/fl^ mice (**Figure 3C**, **Figure 3—figure supplement 1**). Measurement of the

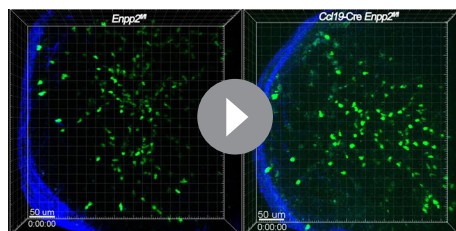

**Video 1.** FRC-specific ATX ablation reduced the velocity of T cells in LNs. Intravital two-photon microscopy was used to analyze the intranodal migration of eGFP-expressing CD4$^+$ T cells (green) 15 hr after the cells were injected into *Enpp2*$^{fl/fl}$ (left) or *Ccl19*-Cre *Enpp2*$^{fl/fl}$ mice (right). Data shown are representative of three independent experiments. Bars, 50 μm.

mean displacement and the motility coefficient, which represents the volume in which an average cell scans per unit time (*Sumen et al., 2004*), also indicated that T-cell motility was impaired in the LN parenchyma of *Ccl19*-Cre *Enpp2*$^{fl/fl}$ mice (*Figure 3D,E*), whereas the directionality of the intranodal T-cell movement was comparable in these mouse groups (*Figure 3—figure supplement 2*), supporting the hypothesis that FRC-derived LPA is required for efficient intranodal T-cell migration.

## Adoptively transferred *Lpar2*-deficient T cells are retained in LNs

To understand how LPA regulates intranodal T-cell migration, we analyzed the LPA-receptor expression in lymphocytes. As shown in *Figure 4A*, CD4$^+$ T cells, CD8$^+$ T cells, and B cells expressed different levels of *Lpar2, Lpar5*, and *Lpar6*, encoding LPA$_2$, LPA$_5$ and LPA$_6$ respectively, with B cells expressing relatively high levels of *Lpar2* compared to T cells.

We next examined whether LPA-receptor deficiency in lymphocytes causes abnormal migration into LNs, as seen in myosin IIA–deficient T cells (*Jacobelli et al., 2010*). Splenocytes from eGFP-expressing mice and from LPA receptor-deficient mice were biotinylated, mixed in equivalent numbers, and injected intravenously into WT recipients, and lymphocytes that migrated into the LNs and spleen were counted 1.5 hr later. As shown in *Figure 4B*, *Lpar5*$^{-/-}$ and *Lpar6*$^{-/-}$ lymphocytes migrated into the LN and spleen at levels comparable to WT lymphocytes. In contrast, there were more *Lpar2*$^{-/-}$ than WT lymphocytes in the LNs but not the spleen (*Figure 4B,C*). Among the migrating cells, this increase was evident for CD4$^+$ T cells and CD8$^+$ T cells but not B cells (*Figure 4D*). A similar increase was also observed when purified CD4$^+$ T cells were used (*Figure 4G*). These results suggest two possibilities for the role of LPA$_2$: one is that LPA$_2$ negatively regulates the entry of T cells into LNs across HEVs, and the other is that LPA$_2$ promotes intranodal T-cell motility and hence their egress from the LNs. To test these hypotheses, we co-injected differentially labeled WT and *Lpar2*$^{-/-}$ T cells and examined their intranodal localization after 90 min by whole-mount LN analysis. Among the donor T cells in the LN parenchyma, *Lpar2*$^{-/-}$ and WT T cells localized in the vicinity of HEVs (within 50 μm distance) at comparable levels (*Figure 4E and F*), indicating that LPA$_2$ plays a minor role in T cell entry into LNs across HEVs. By contrast, *Lpar2*$^{-/-}$ T cells localized more frequently in the area distant (more than 50 μm away) from HEVs compared with WT T cells (*Figure 4E and F*), consistent with the hypothesis that LPA promotes intranodal T cell migration via LPA$_2$, and hence, *Lpar2*-deficient T cells were retained in the LN parenchyma. To directly assess the involvement of the LPA$_2$ signaling in T-cell egress from LNs, we blocked the T cell entry by injecting anti-L-selectin antibody MEL-14 intravenously (T = 0 hr), and measured the number of WT and *Lpar2*$^{-/-}$ CD4$^+$ T cells residing in the LNs 23 hr after the injection. As shown in *Figure 4G*, the blockade of T cell entry (T = 23 hr) resulted in reduction of the cellularity in LNs and increased the ratio of *Lpar2*$^{-/-}$ T cells to WT cells compared to that before the injection. These results indicate that retention of *Lpar2*-deficient T cells was seen even after blockade of lymphocyte ingress to LNs and are compatible with the hypothesis that *Lpar2*-deficiency leads to T cell retention in the LN parenchyma without affecting lymphocyte migration into LNs.

## LPA$_2$-mediated signaling regulates intranodal T-cell migration

To verify that LPA$_2$-mediated signaling is required for intranodal T-cell motility, we next compared the intranodal migration of *Lpar2*-deficient and WT T cells. We adoptively transferred CD4$^+$ T cells from *Lpar2*$^{-/-}$ mice and WT mice and used intravital two-photon microscopy to compare the behavior of these cells in the PLN of the recipient mice. Preliminary experiments indicated that the lymphocyte labeling used for in vivo cell tracking did not affect cell motility under the experimental

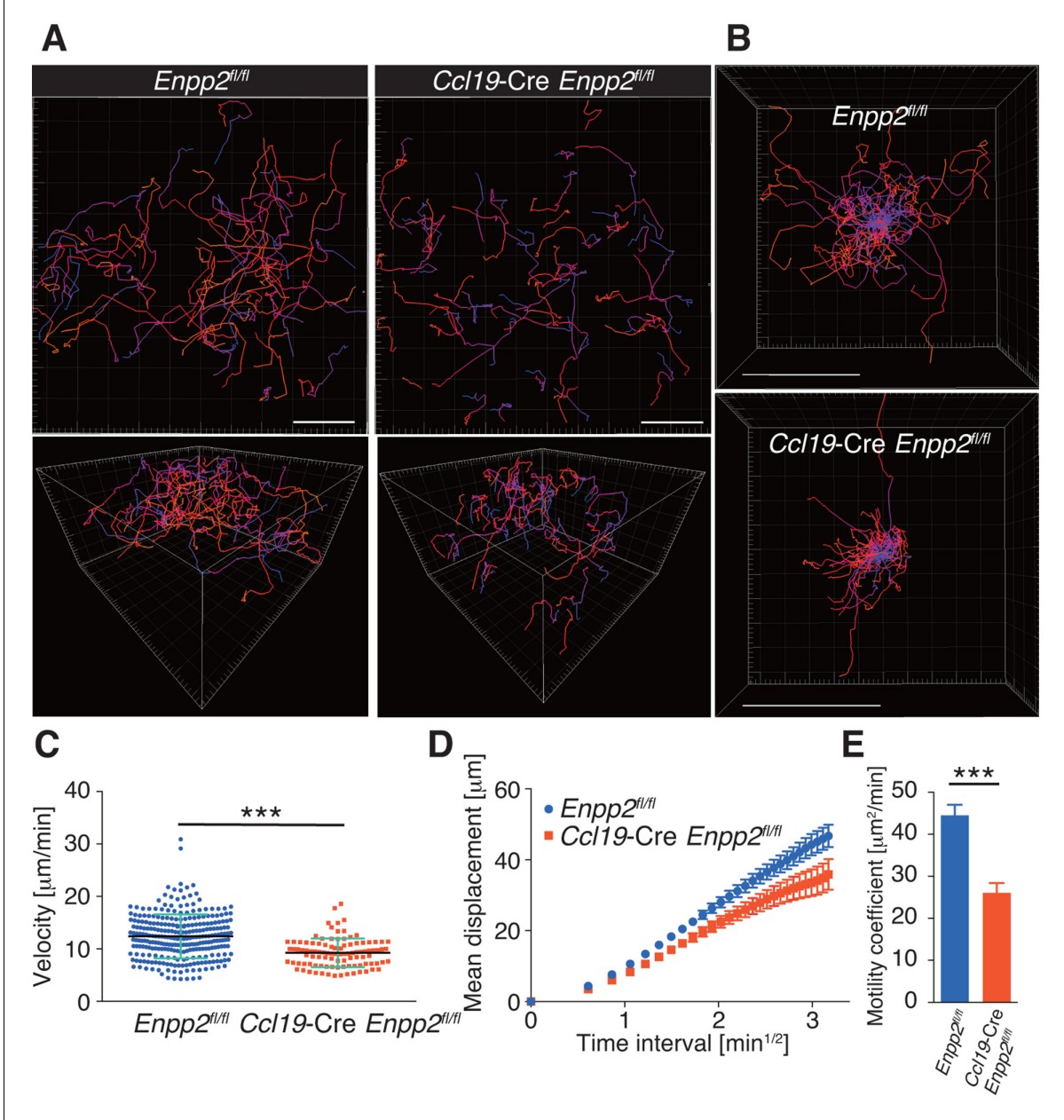

**Figure 3.** FRC-specific ATX ablation attenuates intranodal T-cell migration. Intranodal T-cell migration in *Enpp2*<sup>fl/fl</sup> mice and *Ccl19*-Cre *Enpp2*<sup>fl/fl</sup> mice, analyzed by intravital two-photon microscopy. Intranodal T-cell migration in the PLN was analyzed 15–25 hr after injecting eGFP-expressing CD4$^+$ T cells into *Ccl19*-Cre *Enpp2*<sup>fl/fl</sup> or *Enpp2*<sup>fl/fl</sup> mice. (**A**) Automated tracking of CD4$^+$ T-cell migration (upper panels), and images rotated to display the z-dimension of the volume (lower). Trajectories of 50 randomly chosen cells are displayed as color-coded tracks to show movement over time, from blue (start of imaging) to red (end of imaging). (**B**) Translated tracking with a common origin. (**C-E**) Analysis of T cell motility. See also *Figure 3—figure supplement 2*. (**C**) Average cell velocity. Each dot represents the average velocity of an individual cell, and bars indicate the median. Pooled data are shown in *Figure 3—figure supplement 1*. (**D**) Mean displacement plot. (**E**) Motility coefficients, calculated from the slope of the regression line of the mean displacement plot as $x^2/6t$, where x is the displacement at time t. Data represent the mean ± SD (**C**) or mean ± SEM (**D, E**). Data are representative of three independent experiments. Differences between groups were evaluated by Student's *t*-test. *$P < 0.05$, **$P < 0.005$, ***$P < 0.0005$. Bars, 100 μm.

The following figure supplements are available for figure 3:

**Figure supplement 1.** Pooled data of WT CD4$^+$ T-cell velocity in *Enpp2*<sup>fl/fl</sup> and *Ccl19*-Cre *Enpp2*<sup>fl/fl</sup> LNs.

**Figure supplement 2.** Directionality of WT CD4$^+$ T-cell movement in *Enpp2*<sup>fl/fl</sup> and *Ccl19*-Cre *Enpp2*<sup>fl/fl</sup> LNs.

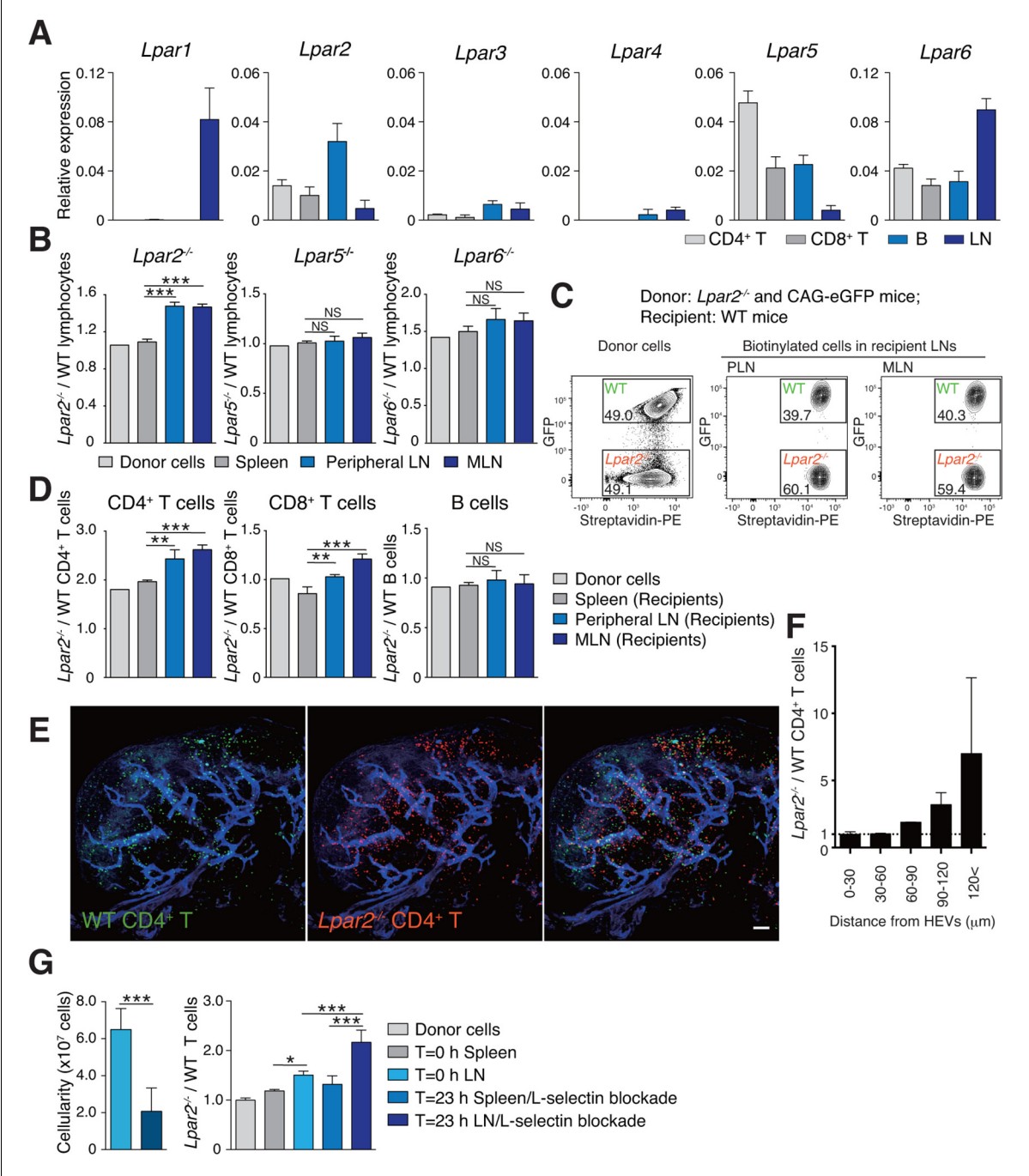

**Figure 4.** *Lpar2* deficiency causes T cells to be retained in LNs. (**A**) LPA-receptor expression in lymphocyte subsets: CD4[+] T, CD8[+]T, B cells and whole LNs were analyzed by real-time RT-PCR; expression levels were normalized to *Gapdh*. Data shown are representative of three individual experiments. (**B–D**) Migration of LPA receptor–deficient lymphocytes into secondary lymphoid tissues. Splenocytes from eGFP-expressing mice and from *Lpar2*[-/-], *Lpar5*[-/-], or *Lpar6*[-/-] mice were mixed in equal numbers, labeled with biotin, and injected intravenously into WT recipients. After 1.5 hr, secondary lymphoid tissues were collected from the recipient mice and donor-derived lymphocytes were detected with flow cytometry. (**B**) Ratio of LPA receptor-deficient (*Lpar2*[-/-], *Lpar5*[-/-] or *Lpar6*[-/-]) cells to eGFP-expressing (WT) cells that migrated into the spleen, peripheral LN, and MLN after adoptive transfer into WT recipient mice. (**C**) Representative dot plots of donor cells (before transfer) and biotinylated cells in LNs of recipient mice (after transfer). (**D**) *Lpar2*[-/-] lymphocyte subpopulations that migrated into the spleen, peripheral LN, and MLN of recipient mice. Data shown for adoptive transfer experiments using *Lpar2*[-/-], *Lpar5*[-/-] and *Lpar6*[-/-] lymphocytes are representative of five, two, and three independent experiments, respectively (n=4 mice per group). (**E, F**) Localization of adoptively transferred WT and *Lpar2*[-/-] CD4[+]T cells in LNs. DiD-labeled CD4[+] T cells from WT mice (1 x 10[7] cells, pseudocolored in green) were mixed equally with CMTMR-labeled *Lpar2*[-/-] CD4[+] T cells (red) and injected intravenously into WT recipient mice; 90 min later, Alexa Fluor 488-conjugated MECA-79 mAb (pseudocolored in blue) was injected intravenously to visualize HEVs. (**E**) LNs were collected and
*Figure 4 continued on next page*

Figure 4 continued

analyzed by confocal microscopy. (F) The distance that CD4[+] T cells migrated from HEVs was measured, and the ratio of *Lpar2[-/-]* to WT T cells was calculated. Data represent mean ± SD from 3 mice. Data are representative of two (E, F) independent experiments. (G) Ratio of *Lpar2[-/-]* T cells to WT T cells retained in LNs when lymphocyte entry into LNs was blocked. CFSE-labeled CD45.1[+] WT and CD45.2[+] *Lpar2[-/-]* CD4[+] T cells were intravenously injected into CD45.2[+] WT mice. Two hours later, anti-L-selectin antibody (MEL-14) was injected (T = 0 hr). At T=0 hr and T=23 hr, the total cell number in LNs (left) and the frequency of donor-derived cells (right) were measured. Data are pooled from two independent experiments. Data represent the mean ± SD (A, B, D, F, G). Differences between groups were evaluated by one-way ANOVA (B, D, G [right]) or Student's *t*-test (G [left]). *$P < 0.05$, **$P < 0.005$, ***$P < 0.0005$. Bars: 100 μm.

conditions (unpublished observation). Compared to WT T cells, the *Lpar2[-/-]* T cells moved noticeably more slowly and had comparatively shorter track lengths and less displacement from their original position per unit of time (*Figure 5A*, *Figure 5—figure supplement 1*, *Video 2*). *Lpar2[-/-]* T cells had a 13% lower average velocity (*Figure 5B*), a 45% lower mean square displacement (*Figure 5C*), and a 70% lower motility coefficient than WT T cells (*Figure 5D*), although the directionality of T-cell motion was comparable in these cell types (*Figure 5—figure supplement 2*). Results using explanted LNs were compatible with those from intravital imaging analysis (unpublished observation).

We next examined the importance of the LPA$_2$ receptor in the FRC-derived LPA-dependent T-cell motility in the LN parenchyma. WT and *Lpar2[-/-]* CD4[+] T cells were mixed and injected intravenously into *Enpp2[fl/fl]* or *Ccl19*-Cre *Enpp2[fl/fl]* mice, and the migration of these cells within the LN was monitored. In *Enpp2[fl/fl]* mice, *Lpar2[-/-]* T cells had significantly less intranodal motility than WT cells (*Figure 5E,G* and *Figure 5—figure supplement 3A,B*). On the other hand, a reduced intranodal motility of *Lpar2[-/-]* T cells was not observed in *Ccl19*-Cre *Enpp2[fl/fl]* mice, which lack ATX/LPA in the FRCs (*Figure 5F,G* and *Figure 5—figure supplement 3A,B*). These results indicate that intranodal T-cell motility is regulated via LPA$_2$ on T cells and that the LPA$_2$-mediated T-cell motility requires FRC-derived LPA.

Because LFA-1 regulates high-speed intranodal T cell migration (*Katakai et al., 2013*; *Woolf et al., 2007*), we examined integrin dependency in the LPA$_2$-mediated intranodal T-cell migration by transferring the WT and *Lpar2[-/-]* T cells and monitoring the cell migration before and after injecting anti-LFA-1 antibody (M17/4). As shown in *Figure 5H–J* and *Figure 5—figure supplement 3C–E*, the blockade of the interaction between LFA-1 and ICAM-1 significantly attenuated WT T cell motility as reported previously (*Katakai et al., 2013*; *Woolf et al., 2007*). Of note, even in the absence of the LFA-1/ICAM-1 interaction, *Lpar2* deficiency reduced T-cell motility, and the extent of reduction in T-cell velocity (19.9 ± 5.5%) and motility coefficient (43.5 ± 16.8%) was comparable to that observed in the presence of the integrin interaction (15.8 ± 4.4% and 43.0 ± 13.6% reduction in velocity and motility coefficient, respectively) (*Figure 5H–J* and *Figure 5—figure supplement 3C–E*). These results support the hypothesis that LPA/LPA$_2$ regulates T-cell motility at least partly in an integrin-independent manner.

## LPA enhances T-cell migration across narrow pores in an LPA$_2$/Rho-dependent manner

LPA is reported to promote T-cell migration in a two-dimensional (2D) environment by inducing chemokinesis (*Katakai et al., 2014*; *Zhang et al., 2012*; *Knowlden et al., 2014*). We therefore investigated the motility of WT and *Lpar2*-deficient CD4[+] T cells in a 2D environment by time-lapse microscopy. Untreated or LPA-pretreated CD4[+] T cells were applied to one side of an EZ-Taxiscan chamber coated with ICAM-1, and the migration of cells toward CCL21 placed on the other side of the chamber was monitored in real time. As shown in *Figure 6A,B*, WT T cells efficiently migrated toward CCL21 in response to LPA in a time-dependent manner. Compared to untreated WT T cells, LPA-sensitized WT T cells had a higher velocity and a smaller mean turning angle (*Figure 6C,D*). In contrast, LPA treatment did not enhance the migration of *Lpar2[-/-]* T cells toward CCL21.

We next examined whether LPA also regulates T-cell migration via LPA$_2$ in a three-dimensional (3D) environment, in which leukocyte migration largely depends on actomyosin contraction (*Lämmermann and Germain, 2014*). Transwell analyses have shown that lymphocyte migration across a filter with a 3-μm pore diameter depends on ROCK/myosin II signaling, while migration

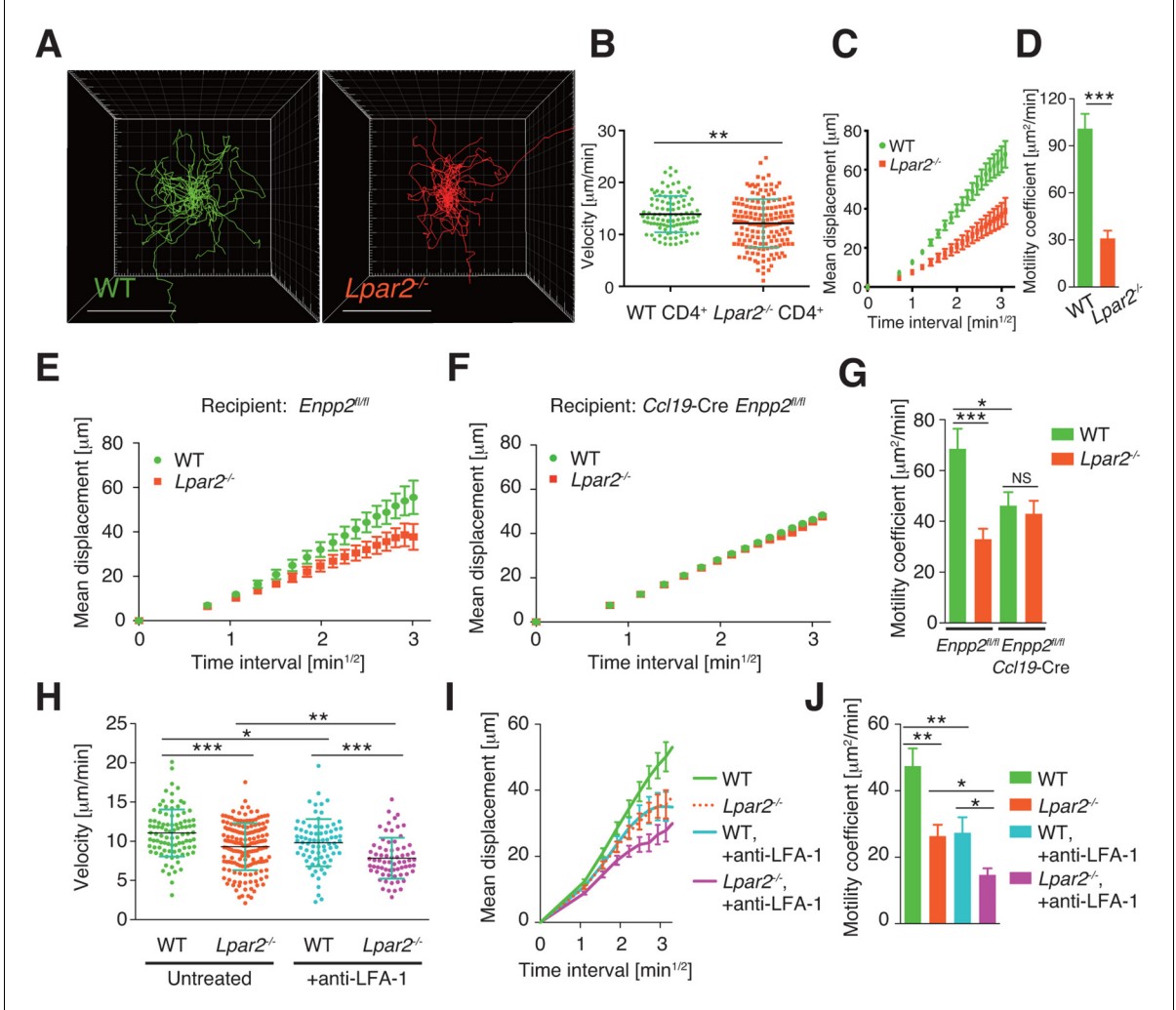

**Figure 5.** *Lpar2* deficiency attenuates intranodal T-cell migration. (**A–D**) Analysis of the intranodal migration of WT and *Lpar2*[-/-] CD4[+] T cells in the PLN of WT mice by intravital two-photon microscopy. CD4[+] T cells from eGFP-expressing mice (green, 5 x 10[6] cells) mixed with an equal number of CMTMR-labeled *Lpar2*[-/-] CD4[+] T cells (red) were injected into WT mice; the transferred cells' internodal migration was analyzed by intravital two-photon microscopy 15–25 hr later (*Figure 5—figure supplement 1*). (**A**) Translated tracking of CD4[+] T-cell migration with a common origin. (**B-D**) Analysis of T cell motility. See also *Figure 5—figure supplement 2*. (**B**) Average cell velocity. Each dot represents the average velocity of an individual cell; bars indicate the median. (**C**) Mean displacement plot. (**D**) Motility coefficient. (**E–G**) The intranodal migration of WT and *Lpar2*[-/-] CD4[+] T cells in the PLN of *Ccl19*-Cre *Enpp2*[fl/fl] or *Enpp2*[fl/fl] mice. *Lpar2*[-/-] CD4[+] T cells and WT CD4[+] T cells were mixed in equivalent numbers and injected into *Enpp2*[fl/fl] (**E**) or *Ccl19*-Cre *Enpp2*[fl/fl] mice (**F**). The mean displacement (**E, F**) and motility coefficient (**G**) are shown. (**H–J**) The intranodal migration of WT and *Lpar2*[-/-] CD4[+] T cells in the presence of anti-LFA-1 blocking antibody. (**H**) Average cell velocity. (**I**) Mean displacement plot. (**J**) Motility coefficient. Data are representative of at least three (**A–D, H–J**) or, two (**E–G**) independent experiments. Pooled data of two photon microscopic analyses are shown in *Figure 5—figure supplement 3*. Data represent the mean ± SD (**B, H**) or mean ± SEM (**C–G, I, J**). Differences between groups were evaluated by Student's *t*-test (**B, D, G, J**) or one-way ANOVA (**H**). *$P < 0.05$, **$P < 0.005$, ***$P < 0.0005$. Bars: 100 μm.

The following figure supplements are available for figure 5:

**Figure supplement 1.** Tracking of WT and *Lpar2*[-/-] T cells in popliteal LN using intravital two-photon microscopy.

**Figure supplement 2.** Directionality of WT and *Lpar2*[-/-] CD4[+] T-cell movement in WT LNs.

**Figure supplement 3.** Pooled data of cell behaviors analyzed by intravital two-photon microscopy.

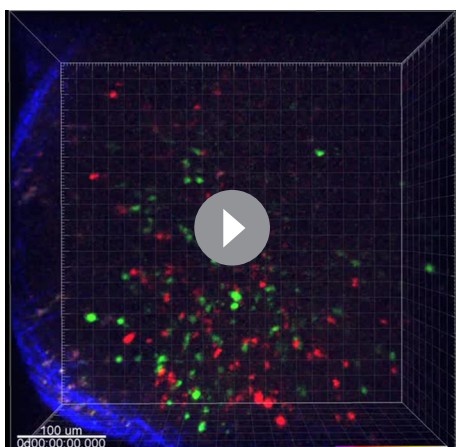

**Video 2.** *Lpar2* deficiency attenuated the intranodal T-cell motility. Intravital two-photon microscopy of the intranodal migration of eGFP-expressing CD4⁺ T cells (green) and CMTMR-labeled *Lpar2⁻ᐟ⁻* CD4⁺ T cells (red) in PLNs 15 hr after the cells were injected into WT mice. Data shown are representative of three independent experiments. Bar, 100 μm.

across a 5-μm pore does not (*Soriano et al., 2011*). Since LPA induces chemokinesis (*Kanda et al., 2008*; *Katakai et al., 2014*), we added LPA to lymphocytes placed in the upper chamber of a Transwell apparatus with a 3-μm pore diameter filter and counted the cells that migrated to the lower well. As shown in *Figure 7A*, LPA sensitization enhanced the migration of CD4⁺ T cells, and this enhancement was strongly inhibited by the ROCK inhibitor Y27632 (*Figure 7A*), or the myosin II ATPase inhibitor blebbistatin (*Figure 7B*), but not by pertussis toxin (PTX), which inhibits Gαi signaling (*Figure 7A*). At 1 μM, LPA prominently induced chemokinesis in WT CD4⁺T cells, consistent with previous reports (*Kanda et al., 2008*; *Katakai et al., 2014*) (*Figure 7C*), but this effect was not observed in *Lpar2⁻ᐟ⁻* CD4⁺ T cells (*Figure 7C*). As shown in *Figure 7D and E*, LPA induced RhoA activation in WT but not *Lpar2⁻ᐟ⁻* CD4⁺ T cells. Reminiscent of the findings of Soriano et al for ROCK/myosin II signaling (*Soriano et al., 2011*), LPA noticeably enhanced the CCL21-dependent migration of T cells across a 3-μm- but not 5-μm-pore diameter filter (*Figure 7F*), and that *Lpar2* deficiency abolished the LPA-induced enhancement (*Figure 7G*).

The myosin II signaling allows T cells to make selective contacts with the adhesive substrates, enabling rapid movement (*Jacobelli et al., 2010*). We thus measured cell surface adhesion area in WT and *Lpar2⁻ᐟ⁻* T cells by total internal reflection fluorescence (TIRF) microscopy (*Jacobelli et al., 2010*). As shown in *Figure 8A,B*, and D, LPA limited T-cell adhesion area on the fibronectin- or ICAM-1-coated substrate in WT T cells, which was inhibited by treatment of cells with blebbistatin, but this effect was not observed in *Lpar2⁻ᐟ⁻* T cells, suggesting that *Lpar2⁻ᐟ⁻* T cells over-adhere to substrate coated with or without LFA-1 ligand. Consistently, the LPA/LPA₂ signaling induced T-cell elongation in a myosin II-dependent manner, allowing swift cell movement (*Figure 8A, C, E*).

We finally analyzed LPA's effect on lymphocyte migration in a more complex confined 3D environment using 3D collagen gel (*Figure 9A*). T cells were placed in type I collagen gel, with LPA placed on one side of the gel and CCL21 plus LPA (5 μM) on the other. WT T cells efficiently migrated toward CCL21 in the presence of LPA, with increased displacement (*Figure 9B*) and velocity (*Figure 9C*) compared with cells stimulated with CCL21 alone, whereas the directionality of T cell movement in the gel was comparable in the presence or absence of LPA (*Figure 9—figure supplement 1*). The LPA's effect was abrogated by treatment of the cells with Y27632 or blebbistatin (*Figure 9D*). In *Lpar2⁻ᐟ⁻* T cells, CCL21 did not enhance motility in the presence of LPA (*Figure 9B, C*). These results collectively indicate that in confined 3D environments, the LPA-LPA₂ axis regulates T-cell motility in a manner dependent on Rho, ROCK, and actomyosin.

Taken together, these results strongly suggest that the LPA₂ signaling modulates T cell motility in the confined environments at least partly in an integrin-independent manner and may also act on cell adhesion to prevent over-adhesion to the substrates. LPA produced by FRCs is essential for this LPA₂-dependent T-cell motility in densely packed LN parenchyma.

## Discussion

LN tissue is densely packed with several types of cells, including lymphocytes, dendritic cells, and mesenchymal stromal cells. Lymphocyte migration in such confined environments is influenced by a delicate balance between actin polymerization/cell adhesion and actomyosin contraction/cell de-adhesion. In this study, we demonstrated that LPA is generated ATX-dependently by FRCs in the LN parenchyma, and that the LPA/LPA₂-mediated signal activates Rho/myosin II in T cells. LPA₂

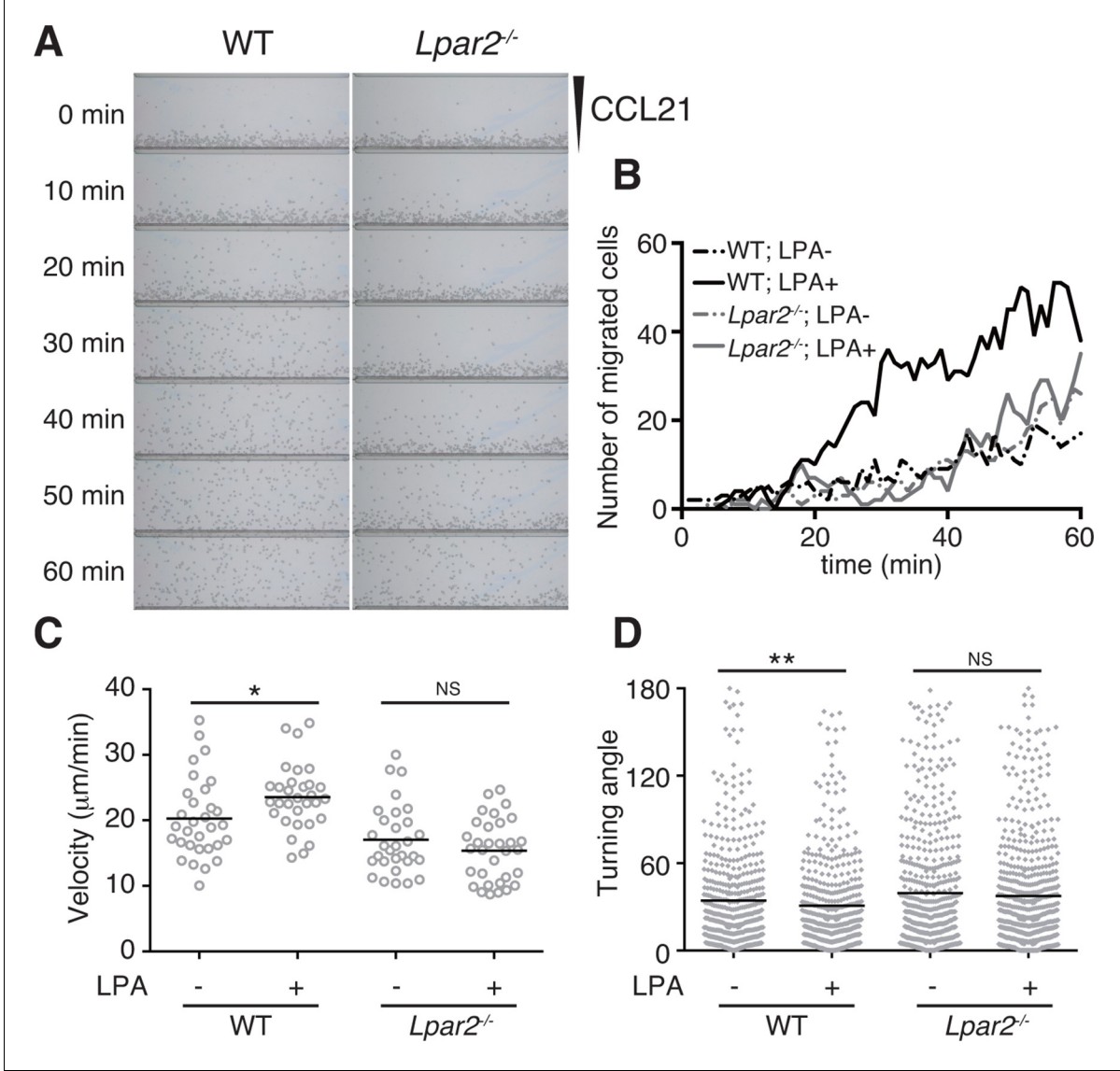

**Figure 6.** LPA/LPA$_2$-mediated signaling promotes T-cell migration on a 2D surface. CD4$^+$ T cells from WT or *Lpar2$^{-/-}$* mice were left untreated or were treated with LPA (1 μM) and immediately loaded on one side of an EZ-Taxiscan chamber, and medium containing CCL21 (100 ng) was applied on the other side. (**A**) Cell migration on a surface coated with ICAM-1-Fc was monitored at 1-min intervals. (**B**) Cells that migrated toward the CCL21-containing contra-wells were counted, and the (**C**) average cell velocity and (**D**) turning angle were calculated. Bars represent the median. Differences between groups were evaluated by Student's *t*-test. *$P < 0.05$, **$P < 0.005$.

signaling enhances T-cell migration in confined 3D collagen matrices and promotes interstitial T-cell migration in LNs in a manner dependent on FRC-derived LPA. These results strongly suggest that the LPA-LPA$_2$ axis critically regulates the confinement-optimized T-cell motility in LNs.

We previously reported that ATX and LPA are produced by HEV ECs and pericytes (*Bai et al., 2013*). In the present study, we found that ATX is also generated by LN FRCs, in agreement with another recent study (*Katakai et al., 2014*), and that multiple LPA species are present in the LN parenchyma; in particular, we showed that LPA (18:1), LPA (18:2), and LPA (20:4) are produced in the paracortex in a manner dependent on FRC-derived ATX. The enzyme ATX has substrate specificity for LPCs with different acyl-chain lengths and saturations. Recombinant mouse ATX prefers LPC species with relatively short, unsaturated acyl chains, and produces LPA (18:2), LPA (16:0), LPA (20:4), and LPA (18:1) but not LPA (18:0) (*Yukiura et al., 2011*). Consistent with this observation, crystal structural analysis revealed that ATX accommodates LPA (18:1) and LPA (18:3) but not LPA

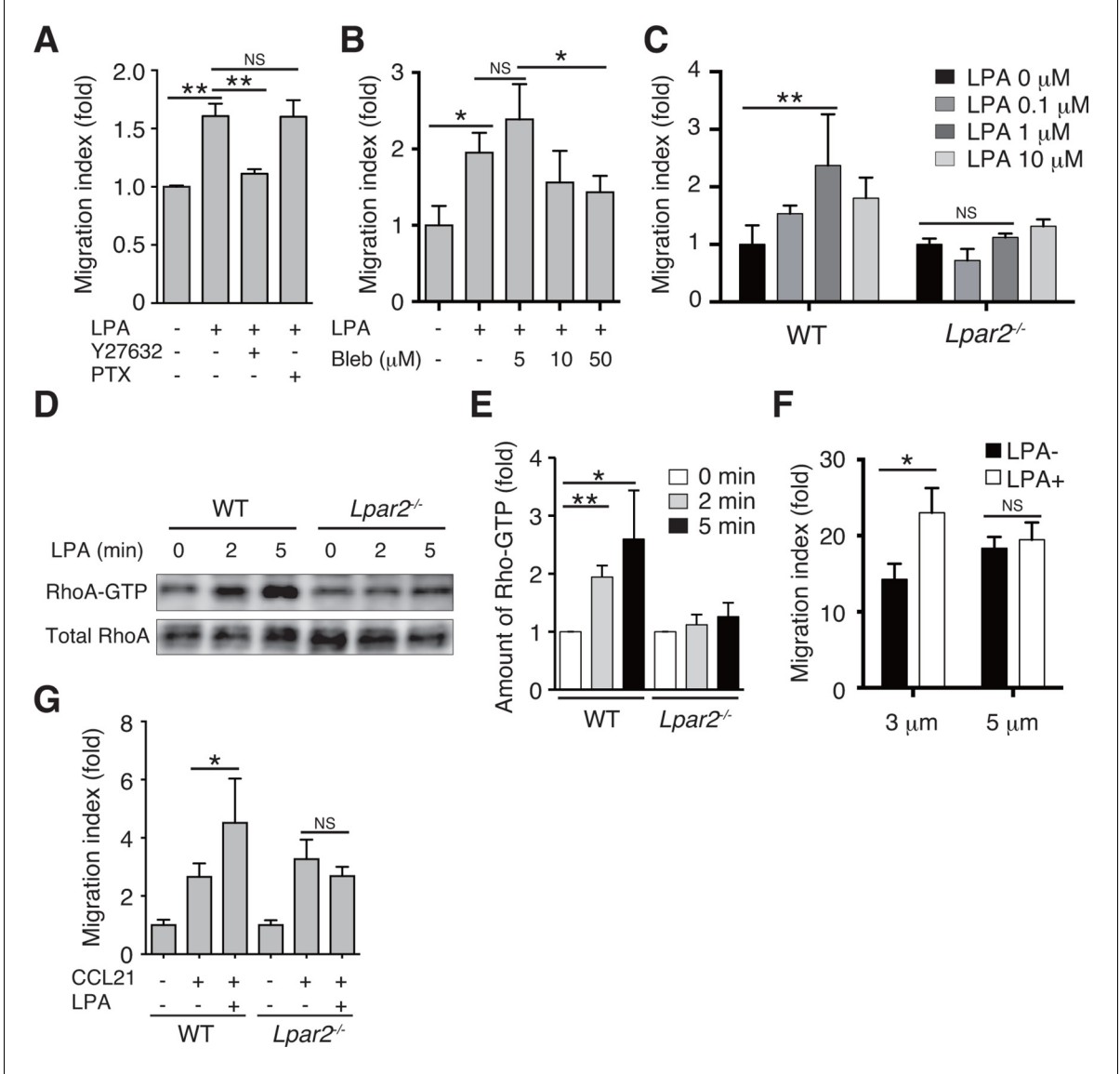

**Figure 7.** LPA/LPA$_2$ axis regulates T-cell migration in a Rho/ROCK/myosin II-dependent manner. (**A, B**) The involvement of ROCK-myosin II signaling in LPA-enhanced cell migration across narrow pores. Lymphocytes were pretreated with the indicated concentrations of blebbistatin (Bleb), Y27632 (10 μM), or PTX (100 ng/ml), and were added with LPA to the upper chamber of a Transwell apparatus with a 3-μm-pore filter. After 2 hr, the migrated CD4$^+$ T cells in the lower chambers were counted by flow cytometry. (**C**) Role of LPA$_2$ in the LPA-induced chemokinesis of CD4$^+$ T cells. WT or *Lpar2$^{-/-}$* lymphocytes were added with various concentrations of LPA to the upper chamber. (**D**) Role of LPA$_2$ in the LPA-induced activation of RhoA. WT and *Lpar2$^{-/-}$* T cells were incubated with 1 μM LPA for the periods indicated, and were lysed immediately thereafter. RhoA-GTP in the lysates was pulled down using Rhotekin and was detected with an anti-RhoA antibody. (**E**) The relative band intensities of LPA-treated WT and *Lpar2$^{-/-}$* T cells, normalized to untreated WT and *Lpar2$^{-/-}$* T cells, respectively. (**F**) Effect of LPA on CD4$^+$ T cell migration through Transwell membranes with a pore diameter of 3 or 5 μm. WT lymphocytes pretreated with or without LPA (1 μM) were applied to the upper chamber, and CCL21 (200 ng/ml) was added to the lower chamber. (**G**) Role of LPA$_2$ in LPA-induced enhancement of CD4$^+$ T cell chemotaxis toward CCL21. WT or *Lpar2$^{-/-}$* lymphocytes were added with LPA (1 μM) to the upper chamber and CCL21 (200 ng/ml) was added to the lower chamber. Data are representative of two (**A, B**) or at least three (**C–G**) independent experiments (n = 3 per group). Data were evaluated by one-way ANOVA (**A–C, F, G**) or Student's t-test (**E**) and represent the mean ± SD. *$P < 0.05$, **$P < 0.005$.

(18:0) in a stable conformation in its hydrophobic pocket, accounting for ATX's preferences for the number of unsaturated bonds of LPC (18:0 < 18:1 < 18:3) (*Nishimasu et al., 2011*). While LPA (18:0) signals were detected in the LN parenchyma, they were ubiquitously found in other tissues independently of ATX expression; these signals are likely to represent intracellularly produced LPA, an

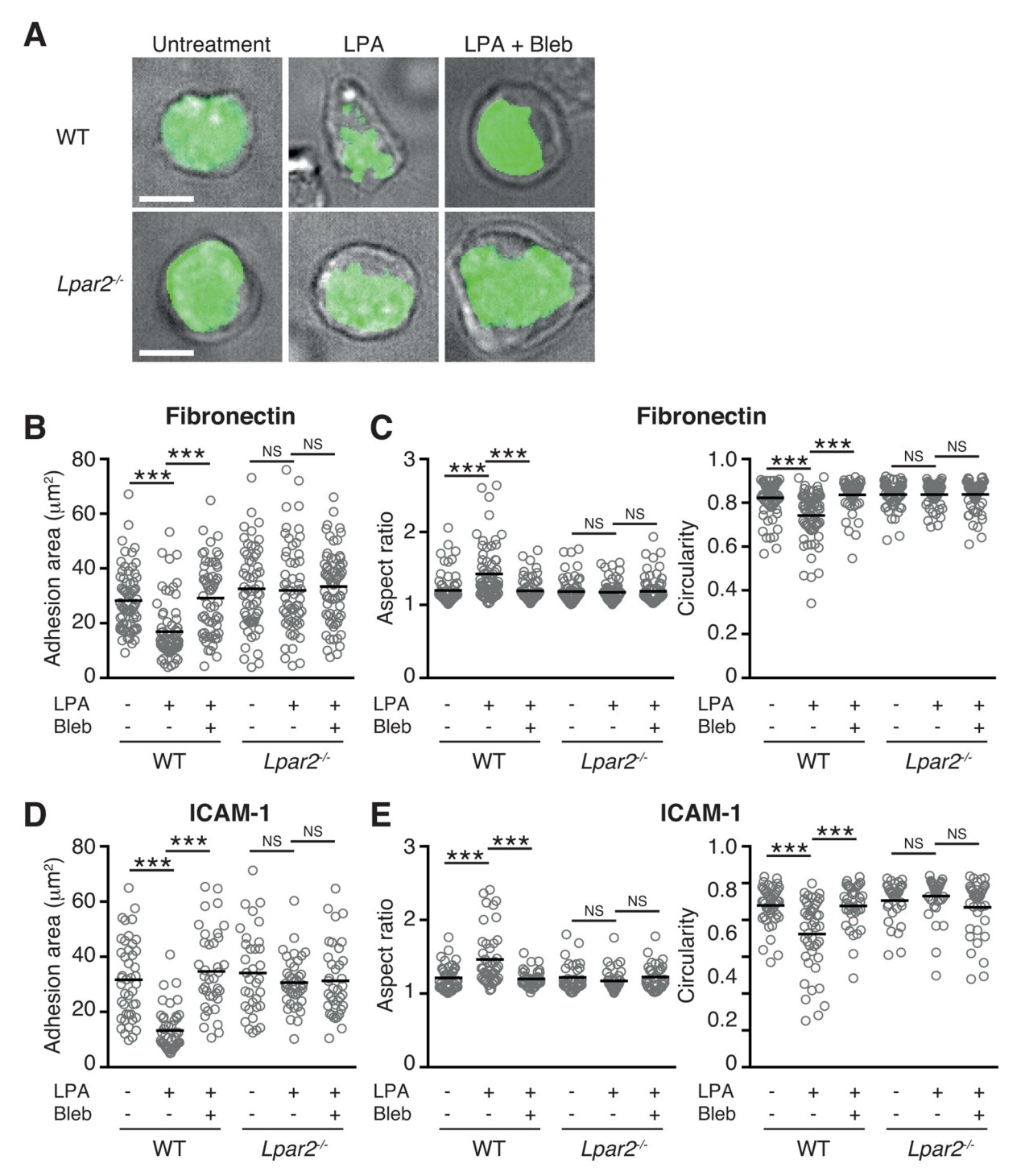

**Figure 8.** LPA/LPA$_2$ axis limits T cell-adhesion area on adhesive substrates. (A–C) Role of the LPA/LPA$_2$ axis in T cell adhesion and morphology. CMFDA-labeled T cells were incubated on a fibronectin- (A–C) or ICAM-1- (D, E) coated slide glass, pretreated with or without blebbistatin (50 μM), and stimulated with LPA (1 μM). The adhesion area of CD4$^+$ cells was assessed by TIRF microscopy (green) (A-B, D). Aspect ratio (cell length/cell breadth) and circularity (4 x $\pi$ x area/perimeter$^2$) (C, E) were assessed by bright-field images. Bars: 5 μm. Data are pooled from two independent experiments. Differences between groups were evaluated by one-way ANOVA. ***$P$ < 0.0005.

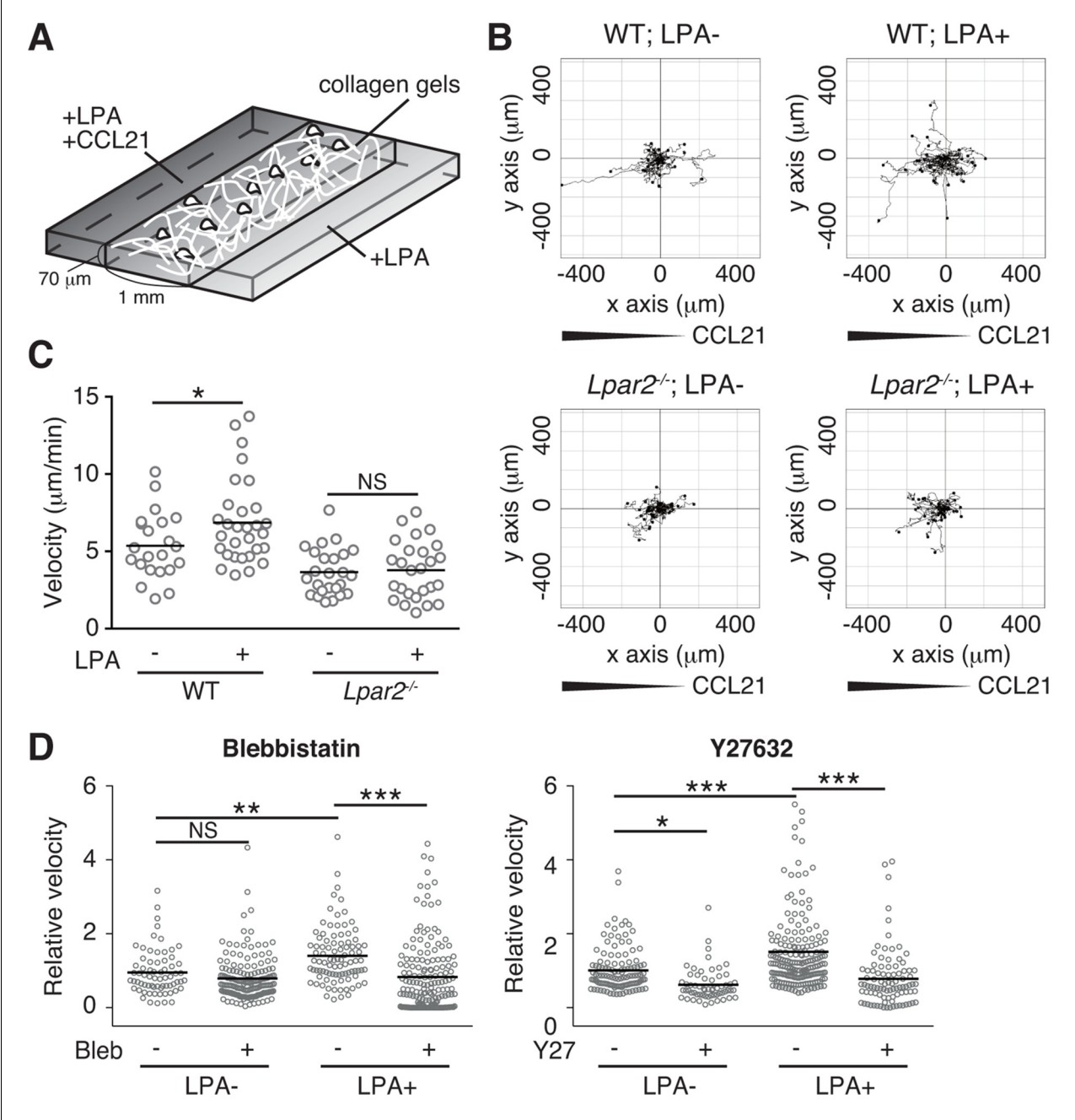

**Figure 9.** LPA/LPA$_2$ axis promotes T-cell motility in a confined 3D environment. (**A**) Illustration of the 3D migration assay. The cell suspension was mixed with collagen gel (1.6 mg/ml). CCL21 (5 μg/ml) was loaded into one side of the gel, and LPA (5 μM) was applied to both sides. The migration of eGFP-expressing CD4$^+$ T cells or their *Lpar2$^{-/-}$* counterparts was recorded for 120 min by time-lapse microscopy. (**B**) Manual tracking of T cell motility. (**C**) T cell velocity. See also the directionality of T cell movement in **Figure 9—figure supplement 1**. (**D**) Involvement of myosin II/ ROCK in LPA-induced T cell motility. After eGFP-expressing CD4$^+$ T cells and WT counterparts were treated with vehicle and indicated inhibitors, respectively, they were mixed with the collagen gel and monitored for cell migration. Data are representative of two independent experiments (**B, C**) and pooled from two independent experiments (**D**). Differences between groups were evaluated by Student's *t*-test (**C**) or one-way ANOVA (**D**). *$P < 0.05$, **$P < 0.005$, ***$P < 0.0005$.

The following figure supplement is available for figure 9:

**Figure supplement 1.** Effect of LPA on the directionality of T-cell movement in 3D collagen gel.

intermediate in the synthesis of triglycerides and glycerophospholipids inside the cell (*Mills and Moolenaar, 2003*), which is relatively inert in activating LPA receptors (*Bandoh et al., 2000*). Taken together, our findings indicate that multiple biologically relevant LPA species (18:1, 18:2, 20:4) are produced by FRCs, which may in turn act locally on LPA receptors expressed by adjacent lymphocytes.

Lymphocyte morphology and migration modes adjust according to the microenvironmental context. T-cell motility on a 2D surface requires integrin-mediated adhesive forces and detachment from the integrin substrates, whereas the 3D T-cell motility in LNs only partially requires LFA-1-mediated interactions (*Woolf et al., 2007*). Jacobelli et al. (*Jacobelli et al., 2010*; *Jacobelli et al., 2009*) reported that lymphocytes are not restricted to adhesion-dependent motility in 3D environments, and that myosin IIA controls T-cell motility modes—including amoeboid motility, which is dependent on the degree of confinement of 3D environment, and prevents excessive adhesions on non-integrin substrates. In our in vitro model, LPA limited T-cell adhesion area on the fibronectin- and ICAM-1-coated substrate, and promoted chemokine-induced T-cell migration across 3-μm but not 5-μm filter pores and also in a confined collagen gel via $LPA_2$ signaling. This enhanced migration was inhibited by the ROCK inhibitor Y27632 or the myosin II ATPase inhibitor blebbistatin. In vivo, the LPA/$LPA_2$ axis promoted intranodal T-cell migration even in the absence of interaction between LFA-1 and its ligand. These findings indicate that LPA acts on $LPA_2$ to enhance T-cell motility in chemokine-rich, confined environments at least partly independently of integrin-mediated adhesion or de-adhesion, and this enhancement depends mainly on ROCK/myosin IIA activity.

The importance of $LPA_2$ in lymphocyte motility is suggested in recent findings by Knowlden et al (*Knowlden et al., 2014*) indicating that *Lpar2* deficiency in lymphocytes is associated with inefficient lymphocyte movement within LNs, although it was not shown how $LPA_2$-signaling affects lymphocyte migration and what kind of cells provide the ligand for $LPA_2$. On the other hand, we showed that LPA activated the Rho GTPase in T cells via $LPA_2$, promoting chemokine-induced T-cell migration across 3-μm- but not 5-μm-pore filters in vitro. We further demonstrated that FRC-derived LPA signaled T cells through $LPA_2$ to sustain optimal T-cell motility in the LN parenchyma. Thus, LPA appears to serve as a cell-extrinsic factor regulating myosin II activity in $CD4^+$ T cells, and the $LPA_2$/ROCK/myosin II signaling pathway is critical for T-cell migration in the confined LN parenchymal environment. In addition, we showed that transferred $Lpar2^{-/-}$ T cells accumulated in LNs more abundantly than WT counterparts 1.5–2 hr after the cell transfer (*Figure 4*), whereas Knowlden et al showed that *Lpar2*-deficiency did not affect T-cell trafficking to LNs significantly 42 hr after cell transfer (*Knowlden et al., 2014*). It has been shown previously that intravenously injected $CD4^+$ T cells swiftly migrate into LNs and a majority of them reside there for only about 12 hr (*Mandl et al., 2012*) before they enter the circulation. Therefore, at the time point used by Knowlden et al (*Knowlden et al., 2014*), labeled T cells in LNs might have included those that have re-entered into LNs.

We and others previously reported that LPA locally produced by HEV ECs regulates constitutive lymphocyte transmigration across HEVs (*Bai et al., 2013*; *Zhang et al., 2012*). This raises the question of whether LPA produced in the HEV area also affects T-cell motility in the LN parenchyma. In this regard, we are currently investigating the effect of EC-specific ATX ablation on lymphocyte behaviors within LNs but do not yet have a definitive answer. Concerning the question of whether LPA produced in the LN parenchyma would affect T-cell extravasation from the HEVs, we found comparable lymphocyte migration across HEVs in $Enpp2^{fl/fl}$ and *Ccl19*-Cre $Enpp2^{fl/fl}$ mice (unpublished observation). Given that *Ccl19*-Cre mice express transgenes in FRCs and in the mesenchymal cells ensheathing HEV ECs, but not in HEV ECs (*Chai et al., 2013*), and that biologically relevant LPA species were produced near HEVs at comparable levels in *Ccl19*-Cre $Enpp2^{fl/fl}$ mice and $Enpp2^{fl/fl}$ mice (*Figure 2B,C*), it appears that ATX produced outside HEVs plays a minor or dispensable role in the lymphocyte transmigration across HEVs.

While B cells expressed high levels of *Lpar2* (*Figure 4A*), adoptively transferred *Lpar2*-deficient B cells were not retained in the LN (*Figure 4D*). Since B cells, like T cells, showed chemokinesis in response to LPA in an $LPA_2$-dependent manner (unpublished observation), B cells may possess functional downstream elements for $LPA_2$-mediated signals. However, whether their intranodal motility depends on motogenic factor(s) other than LPA requires further investigation.

Naïve T cells continuously search for antigen-bearing dendritic cells in the FRC-rich paracortex, and their random migration creates critical opportunities to encounter cognate antigens. It is

possible that LPA promotes immune responses by increasing the T cells' chances of encountering dendritic cells. Once activated by TCR stimulation, T cells reduce their migration speed and $S1P_1$ expression, and thus gain time to proliferate and differentiate into effector cells within the LN (*Schwab and Cyster, 2007*). Interestingly, TCR activation is reported to downregulate $LPA_2$ expression (*Knowlden et al., 2014*; *Huang et al., 2002*), which might reduce the velocity of the T cells and allow them to achieve proper activation.

Taken together, our work offers direct evidence that multiple biologically relevant LPA species produced by FRCs in an ATX-dependent manner act locally on $LPA_2$ on T cells. This $LPA_2$-mediated signaling activates the RhoA-ROCK-myosin II pathway and promotes confinement-optimized interstitial T-cell migration in LNs at least partly in an integrin-independent manner. Thus, the LPA-$LPA_2$ axis is an attractive pharmacological target for novel strategies to control immune responses.

## Materials and methods

### Mice

C57BL/6J mice were obtained from Japan SLC. Beta-actin-eGFP mice were a gift from Dr. Masaru Okabe (Research Institute of Microbial Diseases, Osaka University) (*Okabe et al., 1997*). *Lpar2$^{-/-}$* mice and *Enpp2-flox* mice were provided by Dr. Jerold Chun (the Scripps Research Institute) (*Contos et al., 2002*) and Dr. Woulter H. Moolenaar (The Netherlands Cancer Institute) (*van Meeteren et al., 2006*), respectively, to one of us (JA). The *Ccl19*-Cre mice were described previously (*Chai et al., 2013*). The *Lpar5$^{-/-}$* and *Lpar6$^{-/-}$* mice will be described elsewhere. B6.129X1-*Gt (ROSA)26Sor$^{tm1Hjf}$ (R26*-YFP) mice and CD45.1 mice were purchased from the Jackson Laboratory (Bar Harbor, ME). All mice were housed at the Institute of Experimental Animal Sciences at Osaka University Medical School or Central Animal Laboratory at University of Turku. Animal experiments at Osaka University followed protocols approved by the Ethics Review Committee for Animal Experimentation of Osaka University Graduate School of Medicine. The experiments performed in Finland were approved by the Ethical Committee for Animal Experimentation in Finland and, they were done in accordance with the rules and regulations of the Finnish Act on Animal Experimentation (62/200).

### Reagents and antibodies

Anti-PNAd (MECA-79) mAb was purified using a size-exclusion column with size-exclusion resin (Toyopearl TSK HW55; Tosoh, Japan) from the ascites of mice inoculated with the hybridoma. Purified MECA-79 was labeled with the Alexa Fluor 488 Protein Labeling Kit (Thermo Fisher Scientific, Waltham, MA). APC anti-CD4, Pacific Blue anti-CD8, APC-Cy7 anti-B220, APC anti-gp38, and biotin anti-CD31 mAbs were purchased from Biolegend (San Diego, CA). Anti-L-selectin mAb (MEL-14) was generated from the hybridoma, and anti-integrin LFA-1 (αL) mAb (M17/4) was purchased from Bio X cell (Lebanon, NH). Streptavidin-BD Horizon V500 was obtained from BD Biosciences (San Jose, CA), and LPA (18:1) and blebbistatin were from Sigma-Aldrich (St. Louis, MO). Mouse CCL21 and mouse ICAM-1-Fc were from R&D Systems (Minneapolis, MN).

### Stromal cell isolation

FRCs and BECs were isolated as described previously (*Fletcher et al., 2011*). Cells were incubated with PE anti-CD45, APC anti-gp38, and biotin anti-CD31 mAbs, followed by streptavidin-BD Horizon V500. Stromal-cell populations were sorted by FACSAria (BD Biosciences).

### Real-time RT-PCR

Total RNA was isolated using RNeasy (Qiagen, Germany), and single-strand cDNA was synthesized using M-MLV reverse transcriptase (Promega, Fitchburg, WI) with a random primer. PCR was performed using the Go Taq qPCR system (Promega) at 95°C for 10 min followed by 40 cycles at 95°C for 15 s and 60°C for 1 min. The PCR primer pairs are described in *Supplementary file 1*.

## Transmission electron microscopy

ATX localization was examined by immuno-transmission electron microscopy as previously described (*Bai et al., 2013*). Cryosections were incubated with a guinea pig–derived polyclonal antibody against mouse ATX.

## Soluble LTβR-Fc treatment

The LTβR-Fc expression vector, in which LTβR's extracellular domain was fused to the Fc region of human IgG$_1$, was a gift from Dr. Atsushi Togawa (*Yoshida et al., 2002*) (Fukuoka University Hospital). LTβR-Fc was expressed in 293T cells and purified by a HiTrap Protein A column (GE Healthcare, UK). To block LTβR signaling, LTβR-Fc or control human IgG Fc (100 μg each) was injected intraperitoneally into mice (8–12 weeks old) weekly for 4 weeks as described previously (*Browning et al., 2005*).

## Imaging mass spectrometry (IMS)

IMS was performed as previously described (*Bai et al., 2013*), with minor modifications. Briefly, fluorescein-conjugated dextran (40 kDa) (Thermo Fisher Scientific) was injected into the foreleg footpads to visualize LN structures, and 10 min later, brachial LNs were collected and snap-frozen in liquid nitrogen. Cryosections (8-μm thick) were cut and transferred onto glass slides coated with Indium Tin Oxide (Resistance, 20 ohm; Matsunami, Japan). The sections were washed with 0.03% formate to remove endogenous salts and coated with matrix compound (10 mg/ml 9-aminoacridine dissolved in 70% ethanol). IMS analysis was performed with a MALDI-QIT-TOF-mass spectrometer (iMscope; Shimadzu, Japan) equipped with a 355-nm Nd:YAG laser. Ions at *m/z* 437, 435, 433 and 457, which include ions of LPA (18:0), LPA (18:1), LPA (18:2), and LPA (20:4) respectively, were fragmented by collision-induced dissociation, and the ions produced were analyzed. The ion at *m/z* 153 corresponded to the LPA-specific fragment ion. MS/MS imaging analysis was performed at a raster scan pitch of 50 μm. Image reconstruction of the LPA signals was performed with BioMap Software (Novartis, Switzerland). Images of fluorescein-dextran in the same section were acquired before IMS analysis. For immunohistochemical analyses, serial sections were stained with Alexa Fluor 594-conjugated MECA-79 and Hoechst 33342, and examined by confocal microscopy (LSM710; Zeiss, Germany). Distances between LPA signals and MECA-79[+] HEVs were measured with ImageJ software.

## Intravital two-photon microscopy

Intravital two-photon microscopy was performed as described previously (*Liou et al., 2012*). Briefly, CD4[+] T cells were isolated by negative selection kits (Stemcell Tech, Canada or Miltenyi Biotec, Germany) from the LNs and spleen of eGFP-expressing and *Lpar2*[-/-] mice. *Lpar2*[-/-] CD4[+] T cells were labeled with 5 μM CMTMR (Thermo Fisher Scientific), mixed with an equivalent number of WT CD4[+] T cells, and injected into recipient mice ($5 \times 10^6$ cells/mouse); 15–25 hr later, the mice were anesthetized using isoflurane admixed with O$_2$. The animals were placed on a custom-built preparation stage, the right hind leg was immobilized, and microdissection tweezers were used to expose the PLNs without jeopardizing the blood vessels and afferent lymphatic vessels. Sterile PBS at 37°C was applied to the stage to submerge the LNs, and the temperature was maintained at 37°C using a heating pad with a feedback probe. Two-photon laser-scanning microscopy was performed with an upright Leica TCS SP5 equipped with a 20x water immersion objective (HCX APO, N.A. 1.0; Leica, Germany) and a MaiTai Ti:sapphire-pulsed laser (Spectra-Physics, Santa Clara, CA) set at 880 nm for two-photon excitation. To generate time-lapse series, stacks of more than 30 x-y sections spaced at 2–3 μm were acquired every 20–30 s in resonance mode. For evaluating integrin-dependency in LPA-induced T-cell motility, 300 μg of anti-LFA-1 antibody was intravenously injected, and 1 hr later, cell migration was monitored again. Imaris software (Bitplane, UK) was used to track cells in 3D and measure the cells' velocity and x, y, z coordinates. Mean displacement plots were calculated from the x, y, z coordinates using Excel (Microsoft, Redmond, WA) as described previously (*Sumen et al., 2004*). The motility coefficient (M) was calculated from the mean displacement plot as $M = D^2/6t$, in which D is displacement and t is time, using 10-min tracks. The directionality of cells' motion was calculated by dividing cells' net displacement by the total path length, using over 10-min tracks (*Sumen et al., 2004*).

## Lymphocyte migration assay by flow cytometry

Lymphocyte migration was assayed by flow cytometry as previously described (*Bai et al., 2009*). Splenocytes from eGFP-expressing mice and LPA receptor–deficient mice were mixed in equal numbers and incubated with Sulfo-NHS-LC-biotin reagent (80 μg/ml) (Thermo Fisher Scientific) for 30 min (*Nolte et al., 2004*). The biotin-labeled cells were injected intravenously into recipient mice (2 x 10$^7$ cells/mouse), and the LNs and spleen were harvested 1.5 hr later. The cell suspension was incubated with APC anti-CD4, Pacific Blue anti-CD8, APC-Cy7 anti-B220 mAbs, and streptavidin-PE. The frequencies of T and B cells in GFP$^+$ and GFP$^-$ populations of biotinylated cells were analyzed by flow cytometry (FACSCanto, BD Biosciences).

## Lymphocyte migration assay with whole-mount microscopy

Lymphocyte migration was assayed by whole-mount microscopy as described previously (*Kanemitsu et al., 2005*). Briefly, WT and *Lpar2$^{-/-}$* CD4$^+$ T cells (L-selectin$^+$ CD44$^-$ naïve T cells in CD4$^+$ T cells of WT and *Lpar2$^{-/-}$* mice were 79.7 ± 10.7% and 78.7 ± 9.1%, respectively) were isolated by negative immunomagnetic cell sorting (Miltenyi Biotec) and labeled with 5 μM DiD and 5 μM CMTMR (Thermo Fisher Scientific), respectively. The cells were mixed in equal numbers (1 x 10$^7$ cells) and injected intravenously into WT mice; after 1.5 hr, MECA-79 mAb was injected to label the luminal HEV surface, and inguinal LNs were isolated. The LNs were fixed with 4% paraformaldehyde in phosphate buffer and incubated with sucrose (30%). Immunofluorescence signals were observed with a confocal laser-scanning microscope (FV1000-D, Olympus, Japan). The distance between T cells and the nearest HEV was measured using Imaris software.

## T cell retention assay

T cell retention was assessed as described before with minor modifications (*Pham et al., 2008*). CD45.1$^+$ WT and CD45.2$^+$*Lpar2$^{-/-}$* CD4$^+$ T cells were isolated by negative sorting, mixed in equal numbers (1 x 10$^7$ cells), and labeled with 2 μM of CFSE (Thermo Fisher Scientific). The cells were injected intravenously into recipient WT mice; 2 hr later, the LNs and spleen were collected from half of the recipient mice, and 100 μg of anti-L-selectin antibody (MEL-14) was intraperitoneally injected into the rest of the recipient mice; 23 hr later, tissues were harvested from the antibody-treated mice. The frequency of donor-derived T cells was measured by flow cytometry.

## Time-lapse 2D chemotaxis assay

An EZ-Taxiscan chamber (Effector Cell Institute, Japan) was used for time-lapse chemotaxis assays as described previously (*Bai et al., 2009*). Negatively sorted CD4$^+$ T cells were incubated in serum-free medium for 30 min at 37°C. Cells were suspended in LPA (1 μM)-containing or control buffer (0.1% BSA/RPMI1640) and loaded into microchamber wells (4-μm deep), each holding a cover glass precoated with ICAM-1 (5 μg/ml), and 100 ng of CCL21 was applied to the contra-wells. Phase-contrast images of migrating cells were acquired at 1-min intervals for 1 hr at 37°C. Cells that migrated toward the contra-wells were counted from the images, and 30 migrating cells were tracked using the ImageJ (NIH) manual tracking plug-in. ImageJ software was also used to measure the cells' velocity and x, y coordinates. Turning angles were calculated with Excel using the cells' x, y coordinates.

## Transwell migration assay

LN cells were incubated in serum-free medium (0.1% BSA/RPMI1640) for 30 min at 37°C. The cells (1 x 10$^6$ cells) were added in the presence or absence of LPA to the upper chamber of a Transwell apparatus (pore size, 3 or 5 μm; Corning, Corning, NY) and allowed to migrate for 2 hr at 37°C. In some experiments, CCL21 (200 ng/ml) was added to the lower chamber. Migrated cells were counted by flow cytometry using 6.0-μm Fluoscribe microspheres (Polysciences, Warrington, PA).

## RhoA-GTP pull-down assay

FCS and fatty acid–free BSA were incubated with activated charcoal (Wako, Japan) to remove all lipid fractions. The spleen and LNs were collected, and a single-cell suspension was prepared in RPMI1640 containing 1% FCS. T cells were negatively sorted and starved in serum-free medium (0.1% BSA/RPMI1640) for 2 hr at 37°C. Next, cells (1 x 10$^7$) were stimulated with 1 μM LPA at 37°C

for 2 or 5 min. The cells were collected by centrifugation and immediately dissolved in lysis buffer. The GTP-bound form of RhoA in the lysate was pulled down with Rhotekin and detected by western blotting using the RhoA Activation Assay Biochem Kit (Cytoskeleton, Denver, CO).

## Measurement of T-cell adhesion area and morphology on the substrate

Splenocytes were labeled with 2 μM CMFDA ( Thermo Fisher Scientific) and incubated with biotin-conjugated anti-CD4 mAb. The cells were applied onto human fibronectin (0.002%, Sigma-Aldrich) or mouse ICAM-1-Fc (5 μg/ml)-coated glass slides and starved in serum-free medium (0.1% BSA/RPMI) for 2.5 hr at 37°C. The cells were then cultured in the presence or absence of 50 μM blebbistatin for 15 min, followed by stimulation with LPA (1 μM) for 30 min. After unbound cells were gently washed off, adherent cells were fixed with paraformaldehyde and CD4$^+$ cells were stained by applying Alexa Flour 594-conjugated streptavidin. Surface adhesion zones of CD4$^+$ T cells to the substrate were measured as CMFDA$^+$ area by total internal reflection fluorescence (TIRF) microscopy (*Jacobelli et al., 2010*). TIRF images were acquired with Olympus IX71 equipped with a 100x oil immersion objective (PlanApo; Olympus), 488 nm blue laser (FITEL HPU50101; Furukawa Electric, Japan) and an electron multiplier CCD camera (C9100; Hamamatsu, Japan). The laser and camera were controlled by Metamorph software (Molecular Devices, Sunnyvale, CA). The circularity was calculated as (4 x $\pi$ x area)/perimeter$^2$. The value should range from 0 to 1, and value 1 represents most circular. The aspect ratio was calculated as a ratio of cell length to cell breadth. These morphology parameters were analyzed with brightfield images by using ImageJ software.

## Time-lapse 3D migration assay

3D migration assays were performed with μ-Slide Chemotaxis$^{3D}$ (Ibidi, Germany) according to the manufacturer's protocol. Negatively sorted CD4$^+$ T cells were incubated in serum-free medium for 1 hr at 37°C. In some experiments, the cells were treated by 50 μM blebbistatin or 10 μM Y27632. Cell suspensions were mixed with collagen type I (Advanced Biomatrix, San Diego, CA) and incubated for 1 hr at 37°C to allow collagen polymerization. The final cell and gel concentrations were 9 x 10$^6$ cells/ml and 1.6 mg/ml, respectively. CCL21 (5 μg/ml) and LPA (5 μM) were loaded into one side of the reservoir, and LPA (5 μM) was applied to the other side. Fluorescent and phase-contrast images of migrating cells were acquired at 1-min intervals for 3 hr at 37°C using an inverted microscope (IX-81, Olympus), and 30 migrating cells were manually tracked and their velocity measured using ImageJ software.

## Statistical analysis

Differences between groups were evaluated using Prism software (GraphPad Software, La Jolla, CA) by Student's *t*-test for single comparisons or one-way ANOVA for multiple comparisons, followed by post-hoc Tukey tests. A *P* value < 0.05 was considered significant. Data are presented as mean ± SD unless otherwise described.

## Acknowledgements

We thank T Kondo and Y Magota for technical assistance, and S Yamashita and C Hidaka for secretarial assistance. We also thank Drs. T Tanaka and Y Arima for their critical reading of the manuscript, Dr. Y Okochi for helpful discussion on the experiments, and Dr. Y Okamura for giving us the opportunity to perform TIRF microscopy. This work was supported by Grants-in-Aid 24111005 (to EU and MM) and 24590252 (to EU) from the Ministry of Education, Culture, Sports, Science, and Technology of Japan, and JST PRESTO program (to YS). MS is the leader of JST ERATO Suematsu Gas Biology Project.

# Additional information

## Funding

| Funder | Grant reference number | Author |
|---|---|---|
| Ministry of Education, Culture, Sports, Science, and Technology | 24111005 | Masayuki Miyasaka Eiji Umemoto |
| Ministry of Education, Culture, Sports, Science, and Technology | 24590252 | Eiji Umemoto |
| Japan Science and Technology Agency | | Yuki Sugiura Makoto Suematsu |

The funders had no role in study design, data collection and interpretation, or the decision to submit the work for publication.

## Author contributions

AT, EU, Conception and design, Acquisition of data, Analysis and interpretation of data, Drafting or revising the article; DK, KA, NS, YS, Acquisition of data, Analysis and interpretation of data; HI, KT, NA, JK, Acquisition of data; AI, EH, HH, MS, MI, KT, SJ, Analysis and interpretation of data; ES, SI, Contributed unpublished essential data or reagents; BL, Drafting or revising the article, Contributed unpublished essential data or reagents; JA, Analysis and interpretation of data, Contributed unpublished essential data or reagents; MM, Conception and design, Analysis and interpretation of data, Drafting or revising the article

## Ethics

Animal experimentation: All mice were housed at the Institute of Experimental Animal Sciences at Osaka University Medical School or Central Animal Laboratory at University of Turku. Animal experiments at Osaka University followed protocols approved by the Ethics Review Committee for Animal Experimentation of Osaka University Graduate School of Medicine. The experiments performed in Finland were approved by the Ethical Committee for Animal Experimentation in Finland and, they were done in accordance with the rules and regulations of the Finnish Act on Animal Experimentation (62/200).

# Additional files

## Supplementary files

• Supplementary file 1. Primer sequences for quantitative PCR analysis. These primers were designed using Primer3 software (Sourceforge) and used for SYBR green-based real time-PCR analysis.

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
