## [Decision Letter]

Thank you for submitting your work entitled "Fibroblastic reticular cell-derived lysophosphatidic acid regulates confined intranodal T-cell motility" for peer review at *eLife*. Your submission has been favorably evaluated by Tadatsugu Taniguchi (Senior editor), a Reviewing editor (Johanna Ivaska), and three reviewers.

The reviewers have discussed the reviews with one another and the Reviewing editor has drafted this decision to help you prepare a revised submission.

As you see in the summary based on the reviews, all three reviewers found the work to be interesting and of high quality, as well as potentially suitable for publication, provided you can address the points detailed below. The main point to be addressed is the role of adhesion in LPA-mediated motility and the role of integrin-mediated adhesion and contractility.

Essential revisions:

1) An open question in this study is whether LPA-mediated motility is only required for integrin-independent motility in confined experiments or also for detachment from ICAM-1 through contraction of the adhesive uropod, as reported in other studies (Morin et al., JEM 2008; Soriano et al., JI 2011; Hyun et al., JEM 2012). In fact, the experiments in Figure 6 and Figure 7 appear to support both ideas. One way to address this issue in vivo is to transfer WT and LPA_2_ KO T cells into WT mice, followed by functional blocking of LFA-1 and intravital imaging of lymphoid tissue.

2) The authors should perform static adhesion assays to fibronectin and ICAM-1 to confirm that adhesion is affected in LPA_2_ KO T cells, as suggested by data shown in Figure 8.

3) The authors should show that the LPA-effect on T cell 3D migration in a collagen matrix is dependent on ROCK/myosinII (for example by using blebbistatin and the Rho kinase inhibitor).

4) Knowlden et al. (2014) have reported that LPA_2_ deficiency does not affect steady state T cell homing to lymph nodes. The authors need to compare their results to this published data in the paper and provide an explanation to why they see increased LPA_2_^-/-^ T cell numbers in lymph nodes in their homing assay (e.g. timepoint differences).

5) The interpretation of the egress data in Figure 4/F is highly speculative because distance from HEVs cannot be assigned to an egress phenotype. If the authors wish to conclude that there is difference in retention time, they have to use different experimental setups. For instance, analysis of T cell number after 24 hr (approximately half of CD4^+^ T cells are expected to have exited; Tomura et al. PNAS 2008) combined with CD62L blockade at 2-4 hr post-T cell transfer would be a proper experiment. Alternatively, they may want to re-phrase the text accordingly.

6) On a general note, the authors show too many "representative" experiments. The authors need to pool data (e.g. from intravital imaging or western blotting) in at least one summary graph in addition to detailed graphs of one experiment. For intravital imaging data, information on directionality is missing.

7) Why were splenocytes used in the in vivo migration assay shown in Figure 4? The experiment should be repeated using purified T cells.

[Editors' note: further revisions were requested prior to acceptance, as described below.]

Thank you for resubmitting your work entitled "Fibroblastic reticular cell-derived lysophosphatidic acid regulates confined intranodal T-cell motility" for further consideration at *eLife*. Your revised article has been favorably evaluated by Tadatsugu Taniguchi (Senior editor), Johanna Ivaska (Reviewing editor), and three reviewers. The manuscript is acceptable in principle provided that you address the single remaining point raised by Reviewer #2: the use of the appropriate statistical tests (see below).

*Reviewer #1*:

Convincing revision of a very nice study. I suggest publication.

*Reviewer #2:*

The revised version addresses all concerns in a comprehensive manner and the manuscript reads very well.

*Reviewer #2 (Additional data files and statistical comments)*:

One issue to correct is to use the appropriate statistical test when comparing more than two columns (e.g. in Figure 4, Figure 8 and Figure 9), where the use of Student's t-test is not appropriate.

*Reviewer #3*:

The authors have now addressed the comments I raised. I have no further concerns.

---

## [Author Response]

*As you see in the summary based on the reviews, all three reviewers found the work to be interesting and of high quality, as well as potentially suitable for publication, provided you can address the points detailed below. The main point to be addressed is the role of adhesion in LPA-mediated motility and the role of integrin-mediated adhesion and contractility.*

*Essential revisions: 1) An open question in this study is whether LPA-mediated motility is only required for integrin-independent motility in confined experiments or also for detachment from ICAM-1 through contraction of the adhesive uropod, as reported in other studies (Morin et al., JEM 2008; Soriano et al., JI 2011; Hyun et al., JEM 2012). In fact, the experiments in Figure 6 and Figure 7 appear to support both ideas. One way to address this issue in vivo is to transfer WT and LPA_2_ KO T cells into WT mice, followed by functional blocking of LFA-1 and intravital imaging of lymphoid tissue.*

We thank the reviewers for this valuable comment. Following the reviewers’ suggestion, we intravenously transferred WT and LPA_2_-deficient T cells, and monitored their migration by intravital two-photon microscopy before and after injecting functional blocking antibody against LFA-1. As shown in Figure 5, the blockade of the interaction between LFA-1 and ICAM-1 significantly attenuated the motility of WT T cells as reported by others (Katakai et al., JI 2013; Soriano et al., JI 2011, Woolf et al., Nature Immunol. 2007). In the absence of the integrin-ligand interaction, LPA_2_-deficiency reduced T-cell motility, and the extent of reduction in T-cell velocity (19.9 ± 5.5%) and motility coefficient (43.5 ± 16.8%) was comparable to that observed in the presence of the integrin interaction (15.8 ± 4.4% and 43.0 ± 13.6% reduction in velocity and motility coefficient, respectively). These results are in accordance with the hypothesis that the LPA/LPA_2_ signaling promotes T cell motility in LN at least partly in an integrin-independent manner. We added the description in the last paragraph of the subsection “LPA_2_-mediated signaling regulates intranodal T-cell migration”. As described below, however, the LPA/LPA_2_ axis appears to affect T cell adhesion on the surfaces coated with LFA-1-ligand or fibronectin.

*2) The authors should perform static adhesion assays to fibronectin and ICAM-1 to confirm that adhesion is affected in LPA_2_ KO T cells, as suggested by data shown in*
Figure 8.

We analyzed the adhesion of LPA_2_^-/-^ T cells on the substrate coated with ICAM-1 or fibronectin by TIRF microscopy under static conditions. As shown in Figure 8, LPA limited the cell surface adhesion area of WT cells on the ICAM-1-coated substrate, which was inhibited by the pretreatment of cells with blebbistatin. This LPA’s effect was not observed in LPA_2_^-/-^ T cells, which was also the same with cell adhesion to the fibronectin-coated substrate (Figure 8). We also observed that LPA induced morphological changes in T cells in an LPA_2_-dependent manner on both fibronectin- and ICAM-1-coated surfaces (Figure 8). These data support the hypothesis that LPA_2_ signaling modulates T cell contact with substrates coated with LFA-1 ligand or non-ligand molecule. We described the results in the third paragraph of the subsection “LPA enhances T-cell migration across narrow pores in an LPA_2_/Rho-dependent manner”.

*3) The authors should show that the LPA-effect on T cell 3D migration in a collagen matrix is dependent on ROCK/myosinII (for example by using blebbistatin and the Rho kinase inhibitor).*

Following this comment, we examined the dependency of ROCK/myosin II in the LPA-mediated T cell motility in a collagen matrix and found that LPA-induced T cell motility was abrogated by the treatment of T-cells with blebbistatin or the Rho kinase inhibitor Y27632. These data are included in Figure 9, and the results are described in the fourth paragraph of the subsection “LPA enhances T-cell migration across narrow pores in an LPA_2_/Rho-dependent manner”.

*4) Knowlden et al. (2014) have reported that LPA_2_ deficiency does not affect steady state T cell homing to lymph nodes. The authors need to compare their results to this published data in the paper and provide an explanation to why they see increased LPA_2_^-/-^ T cell numbers in lymph nodes in their homing assay (e.g. timepoint differences).*

We analyzed T cell migration into LNs 1.5-2 h after injecting donor cells, whereas Knowlden et al. performed their analysis 42 h after the injection. It has been reported previously however that, upon intravenous injection, CD4^+^ T cells swiftly migrate into LNs and a majority of them reside within the LNs for only about 12 h (Mandl JN et al. PNAS, 109: 18036-41, 2012) before they enter the circulation again. Therefore, at the time point used by Knowlden et al, labeled T cells in LNs might have included those that have re-entered the LNs. Thus, analyses at earlier time points would reflect the bona fide cell trafficking into LNs more precisely. We discussed this issue in the Discussion section.

*5) The interpretation of the egress data in Figure 4/F is highly speculative because distance from HEVs cannot be assigned to an egress phenotype. If the authors wish to conclude that there is difference in retention time, they have to use different experimental setups. For instance, analysis of T cell number after 24 hr (approximately half of CD4^+^ T cells are expected to have exited; Tomura* et al. *PNAS 2008) combined with CD62L blockade at 2-4 hr post-T cell transfer would be a proper experiment. Alternatively, they may want to re-phrase the text accordingly.*

We appreciate this comment. Following this suggestion, we analyzed WT and LPA_2_-deficient T cell numbers at 23 h after the injection of CD62L antibody into the mice, into which the WT and LPA_2_-deficient T cells were injected 2 h before the antibody administration (T=0 h). As shown in Figure 4, the blockade of T cell entry (T=23 h) increased the ratio of LPA_2_-deficient T cells to WT T cells compared to that observed before injection. These results indicate that retention of LPA_2_^-/-^ T cells was seen even after blockade of lymphocyte ingress to LNs and are compatible with the hypothesis that LPA_2_-deficiency leads to T cell retention in the LN parenchyma without affecting lymphocyte migration into LNs. We have included these data in Figure 4 and described in the Results section (subsection “Adoptively transferred *Lpar2*-deficient T cells are retained in LNs”).

*6) On a general note, the authors show too many "representative" experiments. The authors need to pool data (e.g. from intravital imaging or western blotting) in at least one summary graph in addition to detailed graphs of one experiment. For intravital imaging data, information on directionality is missing.*

In response to the reviewers’ comments, we pooled the data of two-photon microscopic analysis and presented summary graphs in Figure 3—figure supplement 1 and Figure 5—figure supplement 3. We also added the information of directionality in the intravital imaging data (Figure 3—figure supplement 2 and Figure 5—figure supplement 2). In addition, we accumulated the results of Western blotting data and presented them with SD values (Figure 7).

*7) Why were splenocytes used in the* in vivo *migration assay shown in Figure 4? The experiment should be repeated using purified T cells.*

Following the reviewers’ suggestion, we performed the migration assay using purified CD4^+^T cells. As shown in Figure 4, CD4^+^T cells purified from LPA_2_-deficient mice were more frequently found in the WT recipient mice compared to those from WT mice, confirming the importance of LPA_2_ signaling in intranodal T cell motility.

[Editors' note: further revisions were requested prior to acceptance, as described below.]

[…] The manuscript is acceptable in principle provided that you address the single remaining point raised by Reviewer #2: the use of the appropriate statistical tests (see below).

[…]

*Reviewer #2 (Additional data files and statistical comments):One issue to correct is to use the appropriate statistical test when comparing more than two columns (e.g. in Figure 4, Figure 8 and Figure 9), where the use of Student's t-test is not appropriate.* We appreciate this comment. We changed the statistical method for multiple comparisons from t-test to one-way ANOVA followed by post-hoc Tukey tests in Figure 4, Figure 5, Figure 8, Figure 9 and Figure 5—figure supplement 3. We described this information in the Materials and methods section, as well as in the figure legends. This change did not affect the conclusions drawn from the present data.